# OHetTLAL: An Online Transfer Learning Method for Fingerprint-Based Indoor Positioning

**DOI:** 10.3390/s22239044

**Published:** 2022-11-22

**Authors:** Hailu Tesfay Gidey, Xiansheng Guo, Ke Zhong, Lin Li, Yukun Zhang

**Affiliations:** 1Department of Information and Communication Engineering, University of Electronic Science and Technology of China, Chengdu 611731, China; 2Yangtze Delta Region Institute (Quzhou), University of Electronic Science and Technology of China, Quzhou 324000, China

**Keywords:** indoor positioning, online transfer learning, machine learning algorithms, heterogeneous feature spaces, optimization

## Abstract

In an indoor positioning system (IPS), transfer learning (TL) methods are commonly used to predict the location of mobile devices under the assumption that all training instances of the target domain are given in advance. However, this assumption has been criticized for its shortcomings in dealing with the problem of signal distribution variations, especially in a dynamic indoor environment. The reasons are: collecting a sufficient number of training instances is costly, the training instances may arrive online, the feature spaces of the target and source domains may be different, and negative knowledge may be transferred in the case of a redundant source domain. In this work, we proposed an online heterogeneous transfer learning (OHetTLAL) algorithm for IPS-based RSS fingerprinting to improve the positioning performance in the target domain by fusing both source and target domain knowledge. The source domain was refined based on the target domain to avoid negative knowledge transfer. The co-occurrence measure of the feature spaces ($C_{mi}^{p}$) was used to derive the homogeneous new feature spaces, and the features with higher weight values were selected for training the classifier because they could positively affect the location prediction of the target. Thus, the objective function was minimized over the new feature spaces. Extensive experiments were conducted on two real-world scenarios of datasets, and the predictive power of the different modeling techniques were evaluated for predicting the location of a mobile device. The results have revealed that the proposed algorithm outperforms the state-of-the-art methods for fingerprint-based indoor positioning and is found robust to changing environments. Moreover, the proposed algorithm is not only resilient to fluctuating environments but also mitigates the model’s overfitting problem.

## 1. Introduction

With the rapid proliferation of sensor-equipped mobile devices, intelligent applications, and the ubiquity of wireless communication networks, as well as the growing demand for location-based services (LBS), localization, tracking, and navigation systems are receiving much attention from researchers, engineers, practitioners, and the industry [1,2,3]. Location services have greatly facilitated people’s daily activities because the services they need are mostly location-based. Although Global Positioning System (GPS) is the standard method for location determination and is known for its reliable and promising accuracy outdoors, it cannot provide accurate location determination indoors due to non-line-of-sight (NLOS) propagation, poor penetration of GPS signals in complex buildings, and the multipath effect caused by walls and obstacles [4,5,6]. On the other hand, rapid commercial growth and user demand for location-based services and social networking services (SNSs) have made Indoor Positioning Systems (IPSs) essential in all areas of life where mobility plays a role [1,7,8]. Moreover, unlike traditional mobile services, LBSs are used in mobile applications where the user’s current location is needed to provide context-aware functionality [9]. As a result, IPSs are becoming an important research area and are considered a critical element in pervasive computing [10,11]. Indoor positioning is a system for locating mobile devices or objects in a complex indoor environment where GPS signals are attenuated or blocked [3,4].

Various indoor positioning technologies have been proposed in the literature, such as radio-frequency technologies (RFID) [12], Bluetooth (BLE) [13], ultra-wideband (UWB) [14], Zigbee [15]), inertial navigation system [16], Visible Light Communication (VLC) [17], etc., to address the complex nature of the indoor environment, although high implementation costs are required for the additional infrastructure. In the last two decades, various classifications for indoor location technologies have been used [18,19]. In [20], indoor location technologies are classified into building-dependent and building-independent technologies based on the sensors used for localization. According to [20], building-dependent technologies include Wi-Fi, cellular, and Bluetooth, which use the building’s infrastructure, and technologies such as RFID, UWB, infrared, ultrasound, Zigbee, VLC, and acoustic signals, which require their infrastructure. In contrast, image-based technologies and dead reckoning are classified as building-independent. Indoor positioning systems based on wireless networks for building communities, especially Wi-Fi and BLE as they may leverage the building’s infrastructure, have gained significant acceptance from researchers and industry due to their low cost and ease of installation [21,22,23,24,25,26,27]. Along with this, precise information is generally required in modern building domain for indoor localization with the use of BLE network technology including but not limited to emergency management [21], smart plug load control [22], smart HVAC controls [23], route guidance [24], activity recognition [25], monitor occupancy [26], epidemic prevention and health monitoring [27], and others. 

A study in [27], proposes machine learning approaches for a smart contact tracing (SCT) BLE-based system with the goal of epidemic prevention and control. The system used each smartphone’s RSS measurements extracted from the BLE network to estimate its distance from other user’s phones and an alert is issued when social distancing rules are violated. Moreover, the decision tree algorithm has been found to detect potential cases to infectious individuals with an accuracy of 90%, given that two users hold their smartphone in a similar manner. In [21], they have also extracted information from beacons installed in a building which communicate with an occupant’s mobile application for estimating the occupant’s location in an indoor environment (considered both the commercial setting and indoor emergency situation) using machine learning algorithms. In addition, their results [21] revealed that combining BLE with machine learning is certainly promising as the basis for occupancy estimation. It has been also noted that BLE technology has significantly lower power consumption and much lower bit rates than Wi-Fi technology and is ideal for regular short-distance data transmission [28]. Thus, BLE technology is being used or limited for a short length of the advertising packet. Furthermore, privacy security protocols need be preserved to ensure that the BLE’s beacon packet broadcasting will not reveal one’s identity, otherwise scalability of the technology will be negatively affected.

The two well-known indoor localization approaches are the (a) fingerprint-based approach and (b) range-based approach. The range-based approach requires accurate estimates of geometric parameters based on signal features such as received signal strength (RSS) [29], arrival times (TOA) [30], angle of arrival (AOA) [31], and time differences of arrival (TDOA) [32] for the position of the target to obtain a better position estimate. Due to the complex nature of the indoor environment, where radio signals are characterized by the following: NLOS propagation, multipath effects, and a dynamic environment, the distance-based approach cannot provide accurate geometric parameters [4,5,6]. On the other hand, the fingerprint-based approach has attracted much attention in indoor localization because it is independent of geometric parameter estimation and provides better results than the distance-based approach in a complex indoor environment [4,5,6]. Thus, Wi-Fi RSS-based indoor fingerprint positioning (FPBIP) could be implemented without additional infrastructure, as it is widely used for communication purposes [33,34] and is very important for both terminal devices and sensor networks in Internet-of-Things (IoT) applications [35,36]. Therefore, we used Wi-Fi RSS-based fingerprinting, the most popular indoor localization technique [1,33,34,35,36], and included two stages: Offline (training) and Online (localization). In the offline stage, fingerprints are collected from different locations, and the predictive model is trained to learn the relationship between signal and location. The learned model is then used to determine the location of the mobile device or user based on the observed new measurements in the online phase [34,35,36,37].

However, the positioning performance of the FPBIP approach is not robust and efficient due to the dynamic indoor environment. Researchers have proposed several methods to deal with the dynamic indoor environment, which leads to low localization accuracy due to variations in fingerprint patterns over time. These methods can be classified into four groups: probabilistic methods [38,39], machine learning methods [40,41], exploiting the quality of fingerprints of various signal features (such as Channel State Information (CSI) [42], Phase of Arrival (PoA) [43], Time of Flight (ToF) [44], etc.), and derived signal features (Signal Strength Difference (SSD) [15], and Fused Group of Fingerprints [45,46]). Although the proposed methods [38,39,40,41,42,43,44,45,46] have improved the location accuracy, they also suffer from the fluctuations of the signal distribution and are not robust to indoor dynamic environments. This is mainly because (i) the measurements depend on individual fingerprints, (ii) the fingerprint database needs to be up to date, which is impractical, and (iii) a sufficient number of labeled samples are required, which is also very costly. In parallel, hybrid location systems [8,47] (HPS) have been proposed to solve the single location problem, and the results demonstrate better location performance than the single system. The results show better positioning performance than that of a single system. This could be due to the fact that the combination of the two technologies has a leverage effect that compensates for each other’s errors. However, a hybrid base station falls outside the scope of our work and is not economically feasible. On the other hand, computational complexity is a serious problem for hybrid systems based on indoor positioning.

Therefore, it is important to develop a method that reduces the offline calibration effort and still achieves relatively high location accuracy. Transfer learning (TL) methods are effective for many real-world applications to leverage knowledge from the labeled training data of a source domain to improve model performance in a target domain for which there is little or no labeled target training data [48]. However, most TL methods assume that the source and target domains follow the same distribution, so their applications [49] are limited to indoor positioning. Moreover, the labeled training data of the target domain is limited, and it is a difficult task to construct a precise correspondence map to connect the source and target domains. Therefore, we proposed a novel technique to connect the source and target domains to utilize the knowledge of the source and target domains to improve the positioning performance in the target domain. The contributions in this paper are as follows:

(1) We proposed a novel method of construction of a refined source domain (SD) through the feature space transformation technique to enhance quality attributes of the SD in the sense that to avoid negative transfer knowledge to the target domain (TD).

(2) We build a fusing weighting approach to exploit both offline and online instances to enhance the learning performance in the target domain through transfer learning.

(3) Effectiveness of the proposed approach was measured with different machine learning algorithms considering as baselines and verified by conducting extensive experimentations on real-world datasets.

The remaining of the paper is organized as follows: Related works are presented in Section 2. Section 3 described fingerprinting localization framework and its problem formulation. Experimental results, discussions and evaluation metrics are presented in Section 4. Conclusions are provided in Section 5.

## 2. Related Works

There are two main parts to this section: the current state of indoor positioning systems, and the proposed framework of online heterogeneous transfer learning (OHetTL) applied to the indoor positioning problem (IPP).

### 2.1. Indoor Positioning System

Recently, due to the increased demand for LBS and applications, several indoor localization methods have been proposed and implemented [1,2,3,37], but they are not suitable for indoor localization due to some technical limitations and additional infrastructure investment costs [33,34,35,50]. As in [20] indicated, building-dependent technologies include Wi-Fi, cellular, and Bluetooth, which use the building’s infrastructure, and technologies such as RFID, UWB, infrared, ultrasound, Zigbee, VLC, and acoustic signals, which require their infrastructure. In contrast, image-based technologies and dead reckoning are classified as building-independent. Due to their low cost and ease of installation, indoor positioning systems based on wireless networks for building communities, particularly Wi-Fi and BLE as they may utilize the building’s infrastructure, have attracted a lot of attention from researchers and the industry [21,22,23,24,25,26,27]. In [22], they have also described the feasibility of BLE deployment in addressing the plug load management systems to reduce the rising energy consumption of plug loads in commercial buildings through different load monitoring and control strategies. They have further discussed a novel IoT-based occupancy-driven plug load management system called Plug-Mate, which mainly relies on the users’: (a) high-resolution occupancy information obtained through a nonintrusive indoor localization system, (b) plug load type information inferred through an advanced plug load identification feature, and (c) diverse control preferences through a personalized user interface. On the other hand, the authors described the fact that deployment locations of the BLE occupancy sensors are highly dependent on two main determinants: (a) a constant power availability to operate the sensors and (b) communication range of the BLE signal, causing sensors to be set 10–15 m apart from each other and evenly distributed to ensure complete signal coverage [22]. 

Another study in [51] proposed a scalable and less intrusive occupancy detection method-based on smartphones’ device using BLE network technology to perform zone-level occupancy localization, without the need for a mobile application. The results [51] revealed that despite the supervised ensemble model producing the best performance in terms of accuracy and macro average f1-score, the semi supervised clustering model demonstrated practical advantages, as it was able to produce a reasonable performance, while using a fraction of the training data. This could justify that the semi supervised clustering algorithms might be effective in the occupant’s detection for limited labeled samples. Moreover, a non-intrusive occupancy monitoring approach, which leverages on existing BLE technologies found in smartphone devices, has been proposed to track the occupants’ movement patterns using BLE beacons [26]. The proposed approach does not require the installation of a mobile application, just the occupant’s MAC address of their Bluetooth-enabled smartphone devices [26]. Generally, BLE technology has significantly lower power consumption and much lower bit rates than Wi-Fi technology, and is ideal for regular short-distance data transmission [28]. Thus, BLE technology is being used or limited for a short length of advertising packet. In addition to that, intrusive indoor positioning-based BLE technology using the user’s mobile device are becoming more challenging for the scalability or feasibility of deployment due the concern for the user’s privacy. Thus, a privacy security protocol needs be preserved to ensure that the BLE’s beacon packet broadcasting will not reveal one’s identity, otherwise scalability of the technology will be negatively affected. This is consistent with the claims that have been discussed in [52]; regardless, indoor positioning-based BLE network technology is gaining importance due to its ubiquitous nature, low cost, and flexibility, but further scientific developments are limited for novel technologies such as BLE, due to the lack of open datasets and corresponding frameworks suitable to compare and evaluate specialized localization solutions. 

However, Wi-Fi fingerprinting [33,34,35,36,37] is becoming one of the most commonly used methods for indoor localization due to the ubiquity of Wi-Fi infrastructure and the popularity of Wi-Fi-enabled mobile devices. An IPS for commercial buildings has been developed to provide fine-grained occupancy-based HVAC (Heating, Ventilation and Air Conditioning systems (HVAC)) actuation [23]. This system exploits the existing Wi-Fi network infrastructure and occupants’ smartphones with the Wi-Fi enabled. They have also discussed about how using the existing infrastructure for occupancy detection greatly lowers the cost and labor of implementation and maintenance. Furthermore, noisy Wi-Fi signals and metadata regarding the building’s occupants, access points, and HVAC zone have also been employed to alleviate the difficulties in occupancy sensing, which could make it a cost-effective technique. Another method of Wi-Fi indoor localization based on SVM is proposed [53] to identify the floor using the Wi-Fi signal of each floor, and the method achieved an accuracy rate of 99.09% for 3D positioning [53]. An indoor localization system-based Wi-Fi fingerprint using spatial multi-points matching has also been proposed to estimate the user’s position [54]. An indoor location tracking technology-based Wi-Fi fingerprint technique [55] has been proposed to improve the user’s location accuracy using mobile communication technology and location tracking technology. Additionally, for an indoor IoT application using a Bayesian network and a limited radio map, a reliable 3D indoor positioning system-based Wi-Fi RSS fingerprinting has been developed [56]. However, to counteract the impact of received signal variations caused by multipath and signal attenuations throughout various time periods, the radio map should be updated within short time frame.

This method does not use radio signal propagation geometry but requires data acquisition and a built-in radio map in the offline phase, although it has notable shortcomings in the dynamic indoor environment and requires enough labeled samples, making it labor-intensive and costly [57]. Moreover, relatively satisfactory accuracy can be achieved by this method, but [33,34,35] the database fingerprint must be up to date, otherwise, the location performance will be severely degraded. Some research works have been conducted to reduce the effort and time required to create the radio map. Most notably, these include crowd-sourcing [58] and simultaneous localization and mapping over Wi-Fi [59]. However, crowd-sourcing requires the active participation of users or achieves low accuracy, and the Wi-Fi simultaneous localization and mapping approach also suffers from the high computational cost. In [60], it was also described how to speed up computation and enhance system accuracy for indoor environments by using a clustering approach based on fingerprinting. Compared to the non-clustering approach, which had an average distance error of 3.4 m, the suggested technique obtained an average distance error of 2.4 m. Various methods have been proposed in the literature to address the dynamic indoor environment, which results in low localization accuracy due to variations in fingerprint patterns over time. These methods can be classified into four groups: probabilistic methods [38,39], machine learning methods [40,41], using the quality of fingerprints of various signal features (such as Channel State Information (CSI) [3,42], Phase of Arrival (PoA) [43], Time of Flight (ToF) [44], etc.), and derived signal features (Signal Strength Difference (SSD) [15] and fused group of fingerprints [45,46]). 

Although the proposed methods [15,38,39,40,41,42,43,44,45,46] have improved the location accuracy, they also suffer from fluctuations in the signal distribution and are not robust to a dynamic indoor environment, mainly due to two limitations: (i) the measurements depend on a single fingerprint and cannot represent the dynamic scenario of the indoor environment; (ii) the need for a sufficient number of labeled samples is also very costly. Therefore, a hybrid positioning system (HPS) has been proposed for various seamless localization applications (Wi-Fi, Bluetooth, UWB, and ZigBee integrated into a hybrid base station (BS)) to solve the standalone positioning problem [8]. Other researchers in [47] have also proposed a hybrid indoor localization system based on the IMU sensor and smartphone camera, which has better positioning performance than the standalone system due to the possible combination of errors compensated by each technology. However, hybrid BS is not the subject of our work and is not economically feasible. On the other hand, computational complexity is a serious problem for hybrid indoor positioning systems. Research-based on machine learning algorithms (ML) have also been proposed to address the indoor location problem [40,41,61,62]. A limited number of researchers have focused on reducing the calibration overhead for indoor WLAN locations [63,64]. The impact of reducing the number of samples collected at each coverage point, and reducing the number of coverages points on the accuracy of location determination has been studied [63,64]. To compensate for the performance degradation with reduced calibration, both linear and kernel-based interpolation methods have been proposed [63,64] to patch an incomplete radio map. The hidden Markov model [64] and hierarchical Bayesian model [65] have also been proposed to exploit the information contained in the unlabeled user traces and improve the accuracy of location estimation. A label propagation method [66,67] has also been used to learn from labeled and unlabeled data under the assumption that similar data samples should have similar labels. However, this approach is not suitable for online prediction.

### 2.2. Online Transfer Learning Approach to Localization Problem

Several machine learning approaches [61,62] have achieved promising results assuming that the training and testing data have the same distribution. However, in practice, it is not possible to collect enough training data that belong to the same feature space as the test data due to resource and environment constraints. Transfer learning (TL) therefore addresses such cross-domain learning problems by extracting useful information from data in a related domain and transferring it for use in the target tasks. We believe that leveraging the knowledge from the labeled source data can drastically reduce the expensive data labeling effort. Depending on the feature spaces of the source and target domains, TL is divided into two categories: homogeneous transfer (HomTL) and heterogeneous transfer (HetTL). HomTL is used in many real-world applications where the source and target data follow the same distribution, such as text search [68], image classification [69], and face recognition [70]. Although the focus of this work is on IPS, a multiclassification problem that uses binary scores in a source domain has been proposed to alleviate the problem of sparse numerical scores in the target domain [71]. 

Another study highlighting HetTL for multiclass problems was performed based on the compressed sensing theory [72]. In addition, co-occurrence data was used to solve OHetTL problems [73,74,75,76]. A scheme was also explored to evaluate the relatedness between given source domains by transferred weights learned from co-occurrence data [77]. We also found that most existing studies on TL operate in offline batch learning environments, where it is assumed that all training instances of the target domain are given in advance [78,79]. However, this assumption does not hold in many real-world scenarios where collecting sufficient training data is very costly and training instances may arrive online. In the online transfer learning (OTL) problem, we aim to perform an online learning task in a target domain by leveraging knowledge from offline source data and sequentially updating a classifier based on feedback from a data sequence by processing each instance as it arrives [80,81]. 

To address the dynamic indoor environmental conditions, several transfer learning algorithms (TLALs) have been proposed. For example, one study assumed that the RSS value at each location can be computed as a linear aggregate of the RSS values of neighboring benchmarks for each Wi-Fi access point and that this linear relationship remains the same over different time periods [82]. It was found that a model tree was created for each access point at each site. Accordingly, it transmits the Hidden Markov Model (HMM) for a new time point [83]. It uses the neighborhood relation as a bridge to transfer the knowledge and treats the trace as a Markov chain, then the HMM is trained for the new time period. It also considers the problem of cross-device transfer learning and treats multiple devices as multiple learning tasks [84]. However, these algorithms consider either cross-time or cross-device transfer learning and none of them are applied in a real indoor wireless localization system (WILS). The OTL task has also been studied in the context of multi-armed bandits [85]. Moreover, OTL has been applied with online homogeneous source data [86,87]. For online homogeneous TL problems, an ensemble strategy was used, and for OHetTL learning problems, a multi-view approach was used. However, the studies [88,89] considered OHetTL under the assumption that the feature space of the source domain is a subset of the feature space of the target domain. Another study also used some offline data to support an online task in a target domain [88,89,90]. In this sense, the Wi-Fi-based indoor localization system aims to determine the location of the user or mobile device based on the signals received from different Wi-Fi access points (APs). In this work, the proposed model would learn based on the derived new feature spaces refined through the co-occurrence weight score of instances to predict the labeling in the target domain that could arrive in an online (OHetTL). Thus, the Wi-Fi APs send the signal and the mobile user receive the RSS values. The localization process conducted at the user’s side connected to a server. Moreover, Figure 1 below explicitly describes the system operation used for RSS-based Fingerprinting of Indoor Positioning System (FBIPS).

## 3. Problem Formulation and Framework

Consider an indoor environment can be divided into *R* reference points (RP), which represent the target’s location, and each RP is numbered with a label l,(l=1,2,…,L). Suppose there are P detectable Wi-Fi access points (j) in total, and the received signal strengths (RSS) of fingerprints at the *l*th reference point can be represented as a matrix: $M_{l}=\left[r_{l}{}^{J}\left(i\right),r_{l}{}^{J}\left(i\right),\ldots \ldots ,r_{l}{}^{J}\left(i\right)\right]$ where $r_{l}{}^{J}\left(i\right)={\left[r_{l}{}^{1}\left(i\right),r_{l}{}^{2}\left(i\right),\ldots \ldots ,r_{l}{}^{P}\left(i\right)\right]}^{T}$ and represents the *i*th RSS values collected from the *l*th reference point of the $j^{th}$ access point (i=1,2,…,n;j=1,2,…,p). In the OHetTL problem, the target instances $D_{T}=D_{ol}=\left\{X_{T}\right\}={\left\{\left(x_{T}^{m}\right)\right\}}_{m=1}^{n_{T}}\in x$ arrive in an online manner, and the labeled source data $D_{of}=\left\{X_{of},Y_{of}\right\}={\left\{\left(x_{l},y_{l}\right)\right\}}_{i=1}^{n_{of}}\in x_{s}\times y_{s}$ come from an offline source domain, where $n_{T}$ and $n_{of}$ represent the numbers of target and source data, respectively. In the traditional machine learning (ML) techniques, both the training and testing instances be assumed to come from the same domain, and follow the same data distribution. However, in practice, training and testing data may come from different domains, which has either:
Different marginal distributions, or different feature spaces: $X_{S}\neq X_{T},or;P\left(X_{S}\right)\neq P\left(X_{T}\right)$ orDifferent predictive distributions, or different label spaces: $y_{S}\neq y_{T},or,\left(P_{S}\left(y|x\right)\neq P_{T}\left(y|x\right)\right)$.

Several machine learning algorithms have been proposed for indoor positioning problems, but they do not consider the possible effects of signal variations that are likely to originate from the source domain to improve the model’s positioning performance in the target domain. In practice, there may be multiple source domains associated with the target domain that could improve target positioning, although combining these domains is challenging (see Figure 2). The expensive data labeling effort can also be reduced by using the knowledge of the labeled source data. It is also much easier and cheaper to collect unlabeled samples in the field during the testing phase. Therefore, it is important to develop a method that reduces the effort required for offline calibration while still achieving relatively high accuracy in location determination. Figure 2 illustrates the approach of the proposed algorithm, which uses the knowledge from the source and target domains of heterogeneous feature spaces, considering the offline fingerprints of the radio map and the test fingerprints obtained during the online phase as the source and target domains, respectively. First, the proposed algorithm OHetTLAL would derive the new feature spaces based on the weighted co-occurrence of measurements of the offline and online fingerprints, and the newly derived feature spaces are called co-occurrence data. Hoping to improve the learning performance in the target domain, we then fused the offline data and the co-occurrence data to build the offline decision function. Finally, the positioning performance of the model in the target domain is improved by fusing both the offline and online decision functions. In addition, the loss function between the actual and estimated positions was evaluated to update the model. The ⨁ and ⨂ symbols used in Figure 2 represent the data fusion and the updating phase of the classifier based on the loss measurement, respectively. Figure 2 below illustrates the proposed framework of heterogenous transfer learning-based weight co-occurrence for the indoor positioning system.

### 3.1. Distribution Divergence Minimization

In real world scenarios, for the indoor positioning environment, the feature spaces of the sources domain and target domain are different for several practical reasons. Therefore, using the maximum mean discrepancy (MMD) criterion [91,92] distributions of the two domains was compared, and we worked to minimize the distance between the sample mean of the source and target data (reducing the marginal distribution distance of the two domains). To calculate the marginal distribution distance between the source domain and the target domain, the following formula was adopted.
(1)$$Dist\left(X_{s},X_{t}\right)={\left\|\frac{1}{n_{s}}\sum_{i=1}^{n_{s}}x_{s}-\frac{1}{n_{t}}\sum_{m=1}^{n_{t}}x_{t}\right\|}_{F}^{2}$$

### 3.2. Derivation of the New Source Feature Spaces

There are two major problems with OHetTL: (a) the feature spaces of the source and target domains differ. In this manner, the offline database could not be used directly to enhance learning performance, as it could result in a negative knowledge transfer that would degrade positioning prediction accuracy. The labeled training data of the target domain are limited, making it difficult to build a precise correspondence map linking the source and target domains. Thus, our first step in addressing these challenges is to develop new feature spaces that achieve positive knowledge transfer for the target’s localization and are computationally efficient. In order to achieve this, we established a set of standards for constructing new feature spaces called the refined source domain (SD). Exclusion criteria were used to remove redundant knowledge from the offline database during the construction of the new source feature spaces. By excluding irrelevant RSS measurements from different Wi-Fi access points, extraneous features have no longer be included in the analysis. By analyzing the co-occurrence nature of the signal measurements or Wi-Fi access points, this procedure is not only about avoiding or minimizing all negative knowledge transfer to the target domain, but also about identifying the most significant predicting factors.

However, we believe that the similarity between Wi-Fi access points, which is the co-occurrence of the cross-domain Wi-Fi access points, could be used in place of scrutinizing the individual RSS measurements, as the coverage area of a detected access point may approximate the target’s position to some degree. The MMD criterion [91,92] distributions of Wi-Fi access points of the two domains was computed to compare the similarity of those access points (on average) and the motive for this access point similarity would indicate that the coverage area of a detected Wi-Fi access point on average could better approximate the target’s location. Suppose there are P Wi-Fi access points (Wi-Fi APs) detected in both offline and online phases. We define:$J_{of}^{i,p}={\left[J_{of}^{i,1},J_{of}^{i,2},\ldots ,J_{of}^{i,p}\right]}^{T}\&J_{ol}^{m,p}={\left[J_{ol}^{m,1},J_{ol}^{m,2},\ldots ,J_{ol}^{m,p}\right]}^{T}$ to denote the set of detected Wi-Fi APs for offline and online phases, respectively, where $J_{of}^{i,p},J_{ol}^{m,p}\in \left\{0,1\right\}.$ A binary response of the detection indicators would take a value of 1 when the corresponding Wi-Fi APs is detected, otherwise 0 for the undetected, for the *p*th access point in the *i*th offline instance and *m*th online instance, respectively. To construct a refined source domain, a metric was used to compute the co-occurrence between the offline sample and online sample, defined as:(2)$$C\left(i,m\right)=\frac{\sum_{j=1}^{p}\sum_{m=1}^{n_{T}}J_{of}^{i,j}.J_{ol}^{m,j}}{n_{T}}$$

We can calculate the weight co-occurrence scores for all instances of both sides of the source and target domains as: $C_{mi}^{p}=\left(C_{n_{T}n_{S}}^{1},C_{n_{T}n_{S}}^{2},\ldots ,C_{n_{T}n_{S}}^{p}\right).$ The higher C(i,m) would imply that the more detected Wi-Fi APs are shared by these two domains, and the more likely this offline data is useful for the target prediction. The source domain has been refined based on the target domain to avoid negative knowledge transfer. The co-occurrence measures of feature spaces ($C_{mi}^{p}$) were used to derive the homogeneous new feature spaces, and those features with higher weight scores were selected to construct the source domain and used as an input to train the classifier, as it could improve the target’s location prediction. The pseudo code to construct the source domain is provided in Algorithm 1.
**Algorithm 1. Derivation of new source feature space****Input:** (1) Offline sample $D_{of}$; (2) Target domain $D_{T}$; (3) The number of instances $n_{T}$;**Output:** Source domain $D_{S}$1. $D_{S}=\left[\right]$2. **for**
l=1,2,…,L
**do**3.   **for**
m=1,2,…,nT
**do**4.      **for**
$i=1,2,\ldots ,n_{of}$
**do**5.         Compute co-occurrence C(m,i) using Equation (2)6.      **end for**7.     Obtain the co-occurrence vector of the *m*th target sample8.    **end for**9. **end for**10. **for**
$i=1,2,\ldots ,n_{S}$
**do**11.   Find i,m s.t $C\left(m,i\right)=\max C_{mi}^{p}$12.  $x_{of}^{p}=x_{S}^{p^{/}},y_{of}^{p}=y_{S}^{p^{/}}$13.  $D_{S}=\left\{\left(x_{S}^{p^{/}},y_{S}^{p^{/}}\right)\right\}$14. **end for**15. return $D_{S}$

### 3.3. Construction of New Homogeneous Feature Spaces

Consider that there are P Wi-Fi APs detected during offline and online phases as Jofi,p=[Jofi,1,Jofi,2,…,Jofi,p]T&Jolm,p=[Jolm,1,Jolm,2,…,Jolm,p]T respectively, where Jofi,p,Jolm,p∈{0,1}. Let the target RSS measurements arrive in an online manner given as: DT={XT}={(xTm)}m=1nT∈x&Dof={Xof,Yof}={(xl,yl)}i=1nof∈xs×ys for the offline database, respectively. In OHetTL, the source feature space Dof=ℝdof is different from the target feature space DT=ℝdol. The APs shared by the source domain and the target domain can be represented as Jc=(∪i=1nSJSp)∩(∪m=1nTJTp). Assuming that shared APs can offer *dc*-dimensional features, we can select all elements in these specified dimensions to obtain the source and target data with common features, denoted as: XSi,C=[XS1,C,XS2,C,…,XSnS,C]T, and XTm,C=[XT1,C,XT2,C,…,XTnT,C]T respectively, where XSi,C and

XTm,C are in ℝdc. The specific Wi-Fi APs of the source and target domains can be given as: JS(sp)=(∪i=1nSJSp)\Jc and JT(sp)=(∪m=1nTJTp)\Jc respectively, where ‘\’ denotes the set difference operation. The RSS values observed specific to APs of the source and target domains are given as: XSsp=[XS1,sp,XS2,sp,…,XSnS,sp]T, and XTsp=[XT1,sp,XT2,sp,…,XTnT,sp]T respectively, where xSi,sp∈ℝds and xTm,sp∈ℝdT. Thus, the pseudo code for position estimation is provided in Algorithm 2.
**Algorithm 2. The proposed transfer weight optimization and target positioning****Input:** (1). Source domain DS; (2) Target domain DT; (3) Number of instances nT,nS;**Output:** (1). Domain mapping of XS, XT; (2). Weighting transfer Wmi; (3). Target labels yT;1. Initialize **W**2. **for**
l=1,2,…,L
**do**3.   **for**
m=1,2,…,nT
**do**4.      **for**
$i=1,2,\ldots ,n_{of}$
**do**5.         Compute co-occurrence C(m,i) using Equation (2)6.      **end for**7.     Obtain the co-occurrence vector of the *m*th target sample8.    **end for**9. Obtain the common and specific feature matrices of different domains ($J_{S}{}^{\left(sp\right)}$, $J_{T}{}^{\left(sp\right)}$, $J_{C}$)10. **end for**11. **for**
l=1,2,…,L
**do**12.   **for**
$i=1,2,\ldots ,n_{of}$
**do**13.      **for**
$m=1,2,\ldots ,n_{T}$
**do**14.         Compute $W_{mi}$ by using Equations (3)–(7)15.      **end for**16.   **end for**17. **end for**18. Train a classifier from $X_{S\_h}$ and $y_{S}$ by adjusting weights of source samples $W_{mi}$19. Estimate $y_{T}$ on $X_{T\_h}$ by applying the trained classifier $f\left(\left(X_{s\_h},y_{S}\right),W_{mi}\right)$20. **return**
$X_{S}$, $X_{T}$, $y_{T}$, $W_{mi}$

### 3.4. The Objective Function

In OHetTL, the objective function could be affected by several factors, because the environment is inherently heterogeneous. The factors of time period and devices used to collect the signal measurements are not uniform due to the dynamic environment of indoor positioning, and if the data are unbalanced, especially in binary classification, the distribution may be different. Therefore, the proposed solution is to minimize the objective function in a new feature space so that the prior knowledge from the corresponding source domain can be transferred to the target domain. The problem is formulated as a constrained optimization, as can be observed in Equation (3) below. The transformation of the feature spaces in the source domain was based on the attributes in the target domain, where the performance of the classifier helps to improve the prediction of the position of the mobile device or user. Therefore, the new feature spaces were used to minimize the difference in the received signal strength so that the transfer weights can be estimated, as follows: (3)$$\begin{array}{l} \min\sum_{m=1}^{n_{T}}\sum_{i=1}^{n_{S}}w_{mi}{\left\|x_{T\_hm}^{m}-x_{S\_hm}^{i}\right\|}_{2}^{2}\\ s.t.\sum_{i=1}^{n_{S}}e^{-w_{mi}}=1,m=1,2,\ldots ,n_{T} \end{array}$$

The weights constraint is to better minimize the signal measurements of differences between the samples. The equality constraint mentioned in the objective function would assign higher weights to the most related source samples and the lessor weights would be assigned to the less related source samples. The weights are therefore updated as follows. To solve the above constrained optimization problem, Equation (3) can be converted into an unconstrained optimization using the Lagrangian multiplier method. Let us assume that we have obtained a location estimate *z*^(*t*−1)^ at the (*t*−1)th iteration, and now we need to estimate the weights *w*(*t*) at the *t*th iteration. By applying the Lagrangian multiplier method, we obtain:(4)$$L\left(w_{m},\alpha \right)=\sum_{m=1}^{n_{T}}\sum_{i=1}^{n_{S}}w_{mi}{\left\|x_{T\_hm}^{m}-x_{S\_hm}^{i}\right\|}_{2}^{2}+\alpha \left(\sum_{i=1}^{n_{S}}e^{-w_{mi}}-1\right)=0$$
where α is the Lagrangian multiplier. By letting the partial derivative of the Lagrangian with respect to $w_{im}$ and α to be zeros, we obtain:(5)$$\frac{\partial }{\partial w_{im}}L\left(w_{m},\alpha \right)=\left\{ \begin{array}{l} {\left\|x_{T\_hm}^{m}-x_{S\_hm}^{1}\right\|}_{2}^{2}-\alpha e^{-w_{1m}}=0\\ {\left\|x_{T\_hm}^{m}-x_{S\_hm}^{2}\right\|}_{2}^{2}-\alpha e^{-w_{2m}}=0\\ ’\\ ’\\ {\left\|x_{T\_hm}^{m}-x_{S\_hm}^{{}_{n_{s}}}\right\|}_{2}^{2}-\alpha e^{-w_{n_{s}m}}=0\\ \sum_{i=1}^{n_{s}}e^{-w_{im}}-1=0 \end{array}\right.$$

By adding up the first $n_{s}$ terms in Equation (5), we can obtain
(6)$$\alpha =\sum_{i=1}^{n_{s}}{\left\|x_{T\_hm}^{m}-x_{S\_hm}^{i}\right\|}_{2}^{2}$$

Substituting Equation (6) into Equation (5) gives the estimated transfer weight as:(7)$$w_{mi}=-\ln\left(\frac{{\left\|x_{T\_hm}^{m}-x_{S\_hm}^{i}\right\|}_{2}^{2}}{\sum_{i=1}^{n_{s}}{\left\|x_{T\_hm}^{m}-x_{S\_hm}^{i}\right\|}_{2}^{2}}\right)$$

### 3.5. Prediction Evaluation Metrics

In this section, effectiveness of the proposed algorithm was measured with different machine learning algorithms taken as baselines and verified through extensive experimentations based on real world datasets. The root mean square error (RMSE) was used to evaluate the effectiveness of the proposed algorithm and defined as:(8)$$RMSE=\sqrt{\frac{1}{n}\sum_{m=1}^{n}\left[{\left({\hat{x}}_{m}-x\right)}^{2}+{\left({\hat{y}}_{m}-y\right)}^{2}\right]}$$
where ${\left[{\hat{x}}_{m},{\hat{y}}_{m}\right]}^{T}$ and ${\left[x,y\right]}^{T}$ are the predicted location estimate and the true location of a client of the 2-dimensional coordinates of the *m*th positioning sample, respectively. *n* is the total number of samples to be located in the target domain.

## 4. Experimental Results and Discussion

In this section, two real-world experiments were carried out to evaluate our proposed algorithm. Experimental settings and datasets were first presented, and the overall performance of the classifiers are analyzed. Both feature spaces of scenarios were considered (before and after transformation).

### 4.1. Experimental Settings

#### 4.1.1. Dataset A

The experiment was conducted at UESTC (University of Electronic Science and Technology of China) as depicted in Figure 3 and the 21st floor of the Innovation building was taken as the indoor environment site consisting of 10 offices and one corridor covers an area of 1460 m^2^ and partitioned into 210 reference points (RP), which represent the target’s location, and each RP is numbered with a label l,(l=1,2,…,L). $n_{of}$ and $n_{ol}$ are the number of offline and online instances collected during the offline training phases and the online testing phase, respectively. The RSS of fingerprints at the *l*th RP can be represented as a matrix: $M_{l}=\left[r_{l}{}^{J}\left(i\right),r_{l}{}^{J}\left(i\right),\ldots \ldots ,r_{l}{}^{J}\left(i\right)\right].$ Smartphones were used for the data collection at the deployment area.

#### 4.1.2. Dataset B

Another real-office experiment at UESTC (University of Electronic Science and Technology of China) on the 21^st^ floor of the Innovation Building was also conducted in different environmental settings to validate our results, as shown in Figure 4. It covers an area of 1460 m^2^ and partitioned into 175 reference points (RP). Nine Wi-Fi access points were sparsely deployed to guarantee that at least three access points are detectable at each RP.

Moreover, Table 1 below provides the summary of the datasets used to evaluate the algorithms.

### 4.2. Distribution of Homogeneous Feature Spaces

#### 4.2.1. Before Transformation

Table 2 and Table 3 show the distributions of RSS measurements of the offline and online databases, respectively. There were 416 Wi-Fi access points and 15,660 instances were measured from each Wi-Fi access point. The mean RSS values for AP_0, AP_1, AP_2, AP_3, and AP_4 were −59.4, −64.9, −70.1, −74.2, and −73.3, respectively. A standard deviation of 16.8, 15.9, 14.5, 17.9, and 17.1 was also measured for the signal fluctuations. The mean RSS values for AP_412, AP_413, AP_414, and AP_415 are almost all −100, which means that the base stations are not detected at all (Table 2). The unique pattern for these Wi-Fi access points with a score of −100 is that the measured signal variations have a standard deviation of 0.11, 0.43, 0.15, and 0.27, while no signal was detected in any of these specified base stations. The lowest values of standard deviation observed in the above scenario suggest a higher consistency of measurements, even though the standard deviation (sd) can be surprisingly zero for some Wi-Fi access points. However, these base stations (with the lowest sd) were not detected on any mobile device. Such a value (−100) means that no base station was detected during the training and testing phase and we excluded it from the training model. Otherwise, it would be a negative knowledge transfer that either falsely inflate (both false positive and false negative predictions could be high) or degrade the target’s location prediction. Alternatively, the relationship between signal and location might not be characterized.

#### 4.2.2. After Transformation

Table 4 and Table 5 show the distributions of RSS measurements of the derived new features of the source domain and online databases, respectively. Based on the weighted co-occurrence factor, 41 feature spaces were selected to train the classifiers for further location prediction of the target. After creating the new feature spaces, a similar pattern was observed for the Wi-Fi Aps, which had a score of −100, and the signal fluctuations were also inflated to a standard deviation of 16.7, 15.8, 14.5, 17.9 and 14.5 for AP_0, AP_1, AP_2, AP_3, and AP_4, respectively. The higher values of standard deviation that were observed indicate less consistency in the measurements, although the standard deviation may be too large in relative terms for some Wi-Fi access points (with a higher weighting of co-occurrence). Nevertheless, the base stations with a higher weighting of co-occurrence are the ones that have active signal responses to mobile devices in both phases.

Table 6 shows that the number of measurements from each reference point is different and shows that the distribution is unbalanced for dataset A, so feature scaling was considered to avoid the effect of dominance of the larger occurrence of labels within the cluster. Otherwise, the larger features would dominate the others within the cluster. On the other side, the number of measurements collected from each RP demonstrates that the label distribution for dataset B is balanced. Thus, no feature scaling technique is required to avoid the dominance effect of the cluster’s higher label occurrence.

### 4.3. Comparative Analysis of Methods

In this section, we present the comparative analysis of the positioning performance of the proposed indoor positioning-based RSS fingerprinting algorithm. Two real-world datasets were used to compare the proposed algorithm with the most commonly used machine learning algorithms (ML) that have achieved promising results in the literature. Our proposed model was compared with seven machine learning algorithms (Decision Tree (DT), K-nearest neighbor (KNN), Support Vector Machine (SVC), Logistic Regression (LR), Random Forest (RF), Gaussian Mixture Model (GMM), and Neural Network (MLP)). Data were preprocessed, and irrelevant feature spaces were excluded from further analysis based on their weighted co-occurrence scores to avoid negative knowledge transfer. Descriptive analysis was also performed for both datasets of A and B to enable us to provide an insight about the data distributions.

#### 4.3.1. Dataset A

The original number of feature spaces was 416 for dataset A. However, based on the co-occurrence weight scores and exclusion criteria, it was determined that several feature spaces were irrelevant to the analysis because no signal was detected in any of the specified access points. In addition, the Wi-Fi-received signal strength measurements that had the lowest co-occurrence weights for the offline and online source instances were excluded from the prediction task because they did not affect the localization estimate. Figure 5 shows that the RSS measurements for the different Wi-Fi access points have a certain pattern and follow different distributions for the different labels of Wi-Fi AP1 (AP_1). It mainly shows that the relationship between signal and location has been characterized and follows the unique pattern for a particular label. This shows that the signal variations of the received Wi-Fi signal strengths at a particular reference point have been measured with different distribution values, such as the standard deviation and the average values of the received signals. The unique pattern for the measurements collected by the Wi-Fi AP seems to have different values for the signal-to-location relationships of the different target points. In addition to that, there are some noticeable outlier values (as observed in Figure 5 for target points 1, 76, and 191) that may affect the distribution of the values or have a negative impact on the location accuracy (both false positive and false negative predictions could be high) or degrade the location prediction of the target point. Therefore, the relationship between signal and location could not be characterized, as well as the outliers could affect the location performance. Moreover, to better visualize the pattern of the signal distributions in a reference point, some portions of the labels have been extracted, as shown in Figure 6.

Figure 7 shows that the RSS measurements for the different Wi-Fi access points have a certain pattern and follow different distributions for the different labels of Wi-Fi AP4 (AP_4). It shows that the ‘signal-to-location’ relationship has been characterized and follows the unique pattern for the specific label. In other words, the signal variations of the Wi-Fi received signal strengths at a reference point were measured to have different distribution values, including the standard deviation and the average values of the received signals. The unique pattern for the measurements collected by the Wi-Fi AP also appears to have different values for the signal-to-location relationship of the different target points. Moreover, there are some noticeable outlier values (as shown in Figure 8 for target 1) that seem to have extremely fluctuating values at the time of measurement, which could affect the location determination of the target. Therefore, the relationship between signal and location could not be characterized because the outliers could affect the position estimation.

Figure 9 illustrates whether the Wi-Fi signal strength received at a grid point from multiple APs was assumed to be independent such that the Wi-Fi signals transmitted from different APs are transmitted independently and do not interfere with each other. Therefore, the distributions of the RSS measurements for pairs of Wi-Fi access points, especially after transformation techniques were being carried out, show a different pattern of fingerprints for a mobile user at a specific reference point. It shows that the RSS measurements have specific pattern for the different Wi-Fi APs of AP_1 and AP_2 and follows a different distribution of RSS values for the different labels of an access point (dissimilarity). Moreover, the ‘signal-to-location’ relationship was being characterized or studied. The distributions of the RSS values measured at a particular grid point are so dynamic in nature due to the inherent heterogeneity of indoor environment such that there is no guarantee that the signal fluctuations are to be represented by a single value at a specific position.

Figure 10 shows that the RSS measurements have a specific pattern for the different Wi-Fi APs of AP_7 and AP_8 and follows a different distribution of RSS values for the different labels of an access point. Moreover, the ‘signal-to-location’ relationship was being characterized, and the distributions of the RSS values measured at a particular grid point are dynamic in nature due to the inherent heterogeneity of the indoor environment, such that there is no guarantee that the signal fluctuations are represented by a single value at a specific position.

Figure 11a,b show the distributions of RSS measurements of the offline and online databases for dataset A, respectively. There were 416 Wi-Fi access points and 15,660 instances were measured from each Wi-Fi access point; the density distribution of the RSS measurements clearly demonstrate that the mean RSS values were observed to be −100, which means that instances were not received, as the majority of the Wi-Fi APs are not detected for the associated target points. The unique pattern for these Wi-Fi access points with an average score of −100 is that the measured signal variations have the lowest values of standard deviations, while no signal was detected in any of these specified APs. This is what the number tells, nevertheless, it is untrue. However, these APs (with the lowest standard deviation) were not detected on any mobile device during the training and testing phase, and we excluded it from the training model to avoid a negative knowledge transfer that could either falsely inflate (both false positive and false negative predictions could be high) or degrade the target’s position prediction. This is because we cannot talk about the signal distribution of reference points while the associated mobile devices could not receive any signal. Moreover, Figure 11c,d show that the distributions of RSS measurements of the derived new features of the source domain and online databases, respectively. Based on the weighted co-occurrence factor, a new refined source domain was constructed to train the classifiers for the position estimation of the target. One can observe at the density distributions of the refined source domain and target domain that the derived feature spaces are from different distributions with different values of parameters, hence the instances from the offline and online phases are heterogeneous.

Table 7 illustrates the predictive power of different predictive modeling techniques that have been used to evaluate and predict the target location of a mobile device based on the received signal strengths from different Wi-Fi APs. As shown in Table 6 for dataset A, the number of new feature spaces used for the analysis is 41 after the feature transformation was being carried out. In this case, the measurements coming from the remaining Wi-Fi access points were not included in the analysis because they could negatively affect the positioning task. This is because the co-occurrence weights of these Wi-Fi access points from both domains of the instances are insignificant, so including them would falsely inflate our positioning accuracy. Our experimental results demonstrate that the proposed algorithm (OHetTLAL) is the best algorithm with the lowest root mean square error (RMSE). This indicates that our proposed algorithm has succeeded in improving the positioning accuracy by transferring knowledge from the related source domain to the target domain. Not only can a positive knowledge transfer from the source domain to the prediction task be observed, but also the computational cost and memory usage were effectively improved by excluding irrelevant features from the analysis. On the other hand, the Gaussian mixture model has achieved the lowest positioning accuracy and consumes a high computational cost.

Table 8 shows that experiments were also conducted for the indoor practical positioning dataset (Dataset A), and the position estimation accuracy of the different predictive modeling techniques was evaluated to train and estimate the location of a mobile device when Wi-Fi signal strengths were received from different Wi-Fi access points, with the dimensions of the new feature spaces reduced to 15 Wi-Fi access points. The measurements that came from the 26 Wi-Fi access points were also excluded from the analysis because the co-occurrence weight scores of these access points from both domains of the instances were insignificant, so including them would spuriously increase our position accuracy. Similarly, the proposed model was compared with seven machine learning algorithms (decision tree (DT), K nearest-neighbor (KNN), support vector machine (SVC), logistic regression (LR), random forest (RF), Gaussian Mixture Model (GMM), and neural network (MLP)) to see the impact after excluding irrelevant feature domains from the analysis based on their weight scores, hoping to avoid negative knowledge transfer. In addition, those measurements of Wi-Fi received signal strength that had the lowest co-occurrence weight values for the instances of offline and online sources were excluded from the prediction task because they had no impact on the positioning estimation. Moreover, insignificant and redundant features would further affect the computational cost and memory requirements of the algorithm. As shown in Table 8 for dataset A, the number of new feature spaces used for the analysis is 15. Thus, our experimental results demonstrate that the proposed algorithm is the best algorithm with the lowest root mean square error. On the other hand, the Gaussian mixture model (GMM) has obtained the highest error in positioning and consumes a huge computational cost. This indicates that our proposed algorithm has succeeded in improving the positioning accuracy by transferring knowledge from the source domain to the target domain, and that transferring knowledge from the source domain to the prediction task positively affects the positioning accuracy.

Figure 12 confirms that the proposed algorithm is the best algorithm with the lowest root mean squared error for dataset A with 41 feature spaces. In this case, the measurements collected from the remaining Wi-Fi access points were not included in the analysis because they could negatively affect the prediction task. The coincidence weights of these Wi-Fi access points from both domains of instances were found to be insignificant, so including them would spuriously increase the positioning accuracy. All classifiers achieved minimal RMSE even after applying transfer learning. This indicates that our proposed algorithm succeeded in improving the positioning accuracy by transferring knowledge from the corresponding source domain to the target domain. It can also be observed that positive knowledge from the source domain was transferred to the prediction task. On the other hand, the Gaussian mixture model also achieved the lowest positioning accuracy.

Figure 13 shows that the proposed algorithm was the best algorithm with the lowest root mean squared error for dataset A with 15 feature spaces. In this case, the measurements that came from the 26 Wi-Fi access points were also excluded from the analysis because the co-occurrence weight scores of these access points from both domains of the instances were insignificant, so including them would falsely inflate our positioning accuracy. Similarly, all classifiers achieved a minimal RMSE after transfer learning was applied, although slight differences were observed, as some instances were able to falsely improve the RMSE. This indicates that our proposed algorithm succeeded in improving the positioning accuracy by transferring knowledge from related source domains to the target domain. It can be observed that positive knowledge from the source domain was transferred to the prediction task. On the other hand, the Gaussian mixture model also achieved the lowest positioning accuracy.

Figure 14 shows the comparison of the accuracy of indoor positioning classifiers applied to dataset A with new feature spaces with 46 and 15 dimensions. It was found that the proposed algorithm has the lowest root mean square error, i.e., 32% fewer mis-localizations for mobile devices using the 41 new derived features. This indicates that the proposed algorithm has succeeded in improving the positioning accuracy by transferring knowledge from related source domains to the target domain. It can be observed that not only a positive knowledge from the source domain transferred to the prediction task but also the computational cost and memory consumption were effectively improved by excluding irrelevant features from the analysis. However, the Gaussian Mixture Model (GMM) was found to have the lowest accuracy in indoor fingerprint positioning. It leads to mis-localization of mobile devices about 133% of the time and consumes huge computational costs. On the other side, the accuracy of indoor positioning classifiers applied to dataset A with the reduced dimension of 15 features has also been evaluated. It was found that the proposed algorithm has the lowest root mean square error, i.e., predicts 25% fewer wrongly mobile devices. This indicates that the proposed algorithm has succeeded in improving the positioning accuracy by transferring knowledge from related source domains to the target domain. It can be observed that not only positive knowledge from the source domain was transferred to the prediction task, but also the computational cost and memory usage were effectively improved by excluding irrelevant features from the analysis. One can observe the visible differences in positioning accuracy for both scenarios with different dimensions of 41 and 15 feature spaces. Those with 41 feature spaces seem to have better positioning accuracy; however, there are some features whose weight values are almost zero. Moreover, this confirms the claim that the measurements coming from the 26 access points that were excluded from the analysis due to the weighting values for the co-occurrence of these access points from both ranges of instances were insignificant, so the inclusion of these instances would falsely increase our positioning accuracy. On the other hand, the Gaussian mixture model (GMM) was found to have the lowest accuracy in locating indoor objects based on fingerprints. In about 137% of the cases, mobile devices are predicted incorrectly and high computational costs are incurred.

Figure 15 illustrates the impact of different feature spaces used to train indoor positioning classifiers using dataset A with dimensions of 41 and 15, respectively. It was found that the proposed algorithm has the lowest root mean square error, i.e., 32% and 25% less error in positioning the mobile device for both scenarios. This indicates that the proposed algorithm has succeeded in improving the positioning accuracy by transferring knowledge from the source domain to the target domain. It can be observed that not only positive knowledge from the source domain was transferred to the prediction task, but also the computational cost and memory consumption were effectively improved by excluding irrelevant features from the analysis. The difference in positioning accuracy between the two scenarios is about 7%. This difference led us to the important finding that all classifiers achieved lower RMSE both before and after transfer learning for the new feature spaces of dimension 41. However, the contributions of these features to target prediction were not as significant (since their co-occurrence weights of those 26 features were almost negligible) as for the feature spaces with dimension 15. This could show us researchers that a model with a lower RMSE does not mean that it is always the most consistent model, but that we need to examine the parameters for their negative effects on target prediction.

#### 4.3.2. Dataset B

The original number of feature spaces was nine for dataset B. Figure 16 illustrates the distributions of the received signal strength measurements for a given Wi-Fi access point with their corresponding target values. It shows that the RSS measurements have a specific pattern for the different Wi-Fi access points and follow a different distribution of RSS values for the different labels of Wi-Fi AP one (AP_1) of dataset B. This shows that the signal variations of the Wi-Fi-received signal strengths were measured at a reference point and have different distribution values, including the standard deviation and the average value of the received signals. The unique pattern for the measurements collected from Wi-Fi AP (AP_1) also appears to have different values for the signal-to-location relationships of the different target points. Beyond that, there are no noticeable outlier values (as can be observed in Figure 15, except for the RSS values of −90, which demonstrate that the corresponding access points were not detected by any mobile devices).

Figure 17 shows that the RSS measurements have a specific pattern for the different Wi-Fi access points and follow a different distribution of RSS values for the different labels of Wi-Fi AP_3 of dataset B. This shows that the signal variations of the Wi-Fi-received signal strengths were measured at a reference point and have different distribution values including the standard deviation and the average values of the received signals. The unique pattern for the measurements collected by the Wi-Fi AP_3 also appears to have different values for the relationships between the signal locations of the different target points. In addition, there are no noticeable outlier values (as can be observed in Figure 16, except for the RSS values of −90, which demonstrate that the corresponding APs were not detected by any mobile devices).

Figure 18 shows that the RSS measurements for the different pair of access points have a certain pattern and follow the different distribution of RSS values for the different labels of an access point (dissimilarity). It is further described that the Wi-Fi signal strength received at a network point by multiple APs is assumed to be independent so that the Wi-Fi signals sent by different APs are transmitted independently and do not interfere with each other. Therefore, the distributions of RSS measurements for pairs of Wi-Fi access points demonstrate a different pattern of fingerprint for a mobile device at a given reference point even after performing transformation techniques. It can be observed that the RSS measurements for the different Wi-Fi access points of AP_3 and AP_7 have a specific pattern and follow the different distribution of RSS values for the different labels of Wi-Fi access points (dissimilarity). Thus, the relationship between signal and location has been characterized or studied and shows that the distributions of RSS values measured at a given network point are dynamic due to the inherent heterogeneity of the indoor environment. Thus, there is no guarantee that the signal variations at a particular location are represented by a single value.

Figure 19 describes whether the Wi-Fi signal strength received at a grid point from multiple APs can be assumed to be independent. Thus, the Wi-Fi signals transmitted by different APs are transmitted independently and do not interfere with each other. Therefore, the distributions of RSS measurements for pairs of Wi-Fi APs still follow a different pattern of fingerprint distribution for a mobile user at a given reference point after applying transformation techniques. It also shows that the RSS measurements for the different Wi-Fi access points of AP_8 and AP_9 have distinct patterns and follow the different distribution of RSS values for the different labels of a Wi-Fi access point. The “signal-to-location” relationship was established and further conveys that the distributions of RSS values measured at a given grid point are dynamic, and therefore there is no guarantee that the signal variations are represented by a single value at a given location.

Table 9 shows extensive experiments applied to real-world indoor positioning scenarios using dataset B. The original number of feature spaces was nine for dataset B. Nevertheless, experimental results confirmed that the proposed algorithm is the best algorithm with the minimum root mean square error for dataset B, regardless of which feature spaces were fully considered in the analysis. Since the number of feature spaces for dataset B is very small, we could not see or feel the actual effect of irrelevant features. However, based on our given scenarios, some redundant features need to be handled to minimize both computational cost and memory requirements. Experimental results have demonstrated that all classifiers achieved a minimum RMSE after applying transfer learning. This indicates that our proposed algorithm has succeeded in improving the positioning accuracy by transferring knowledge from the source domain to the target domain. However, there is no clear evidence against the fact that negative knowledge was not transferred from the source domain to the target domain. In this case, the measurements that came from all Wi-Fi APs were not refined or reviewed to determine whether or not they should be used for location analysis, because they could negatively affect the location process. On the other hand, the Gaussian mixed model also achieved the lowest location accuracy and consumed higher computational costs.

Table 10 shows detailed experiments for real-world indoor positioning scenarios using dataset B. Here, we evaluated the positioning accuracy of the different predictive modeling techniques to estimate the target position of a mobile device based on the received signal strengths from different Wi-Fi APs. The proposed model was compared with seven machine learning algorithms. The data were preprocessed and irrelevant feature spaces were excluded from further analysis based on their weighted co-occurrence score to avoid negative knowledge transfer. The original number of feature spaces was nine for dataset B before feature transformation was being carried out. However, based on the co-occurrence weight scores and exclusion criteria, three feature spaces were deemed irrelevant for analysis because the co-occurrence weight scores were zero in each of the specified base stations. In addition, the Wi-Fi-received signal strength measurements that achieved the lowest co-occurrence weight scores for the offline and online source instances were excluded from the prediction task because they would have no effect on the position estimation. Accordingly, as shown in Table 9 for dataset B, only six Wi-Fi APs are used as new feature spaces to train the classifiers. In this case, the measurements that came from the remaining Wi-Fi APs were not included in the analysis because they could have a negative impact on the positioning process, since the co-occurrence weight scores of these APs from both domains of the instances were insignificant, and therefore including them would falsely inflate our positioning accuracy. The experimental results confirm that the proposed algorithm is also the best algorithm for dataset B with the lowest root mean square error. This indicates that the proposed algorithm has succeeded in improving the positioning accuracy by transferring knowledge from related source domains to the target domain. It can be observed that not only a positive knowledge was transferred from the source domain to the prediction task, but also the computational cost and memory usage were effectively improved by excluding irrelevant features from the analysis. On the other hand, the Gaussian mixture model has achieved the lowest positioning accuracy and consumes a high computational cost.

Figure 20 shows that the proposed algorithm is the best algorithm with the lowest mean square error for dataset B, with nine Wi-Fi APs. On the other hand, the Gaussian mixture model has also achieved the lowest positioning accuracy. All classifiers have achieved minimum RMSE after applying transfer learning. This indicates that our proposed algorithm has succeeded in improving the positioning accuracy by transferring knowledge from the source domain to the target domain. However, there is no strong evidence against the fact that a negative knowledge was not transferred from the source domain to the target domain. In this case, the measurements coming from all access points were not refined or analyzed to determine whether or not they should be used for location analysis because they could negatively impact the positioning process.

Figure 21 shows that the proposed algorithm was the best algorithm with the lowest mean square error for dataset B with six feature spaces. In this case, the measurements coming from the three Wi-Fi APs were not included in the analysis because they could negatively affect the prediction task. This is because the coincidence weights of these Wi-Fi access points from both domains are close to zero, so including them would affect the positioning performance. Similarly, all classifiers achieved minimum RMSE after applying transfer learning. This indicates that our proposed algorithm has succeeded in improving the positioning performance by transferring knowledge from the corresponding source domain to the target domain. It can be observed that positive knowledge from the source domain was transferred to the prediction task. On the other hand, the Gaussian mixture model also achieved the lowest positioning accuracy.

Figure 22 describes the comparison of the accuracy of indoor positioning classifiers applied to dataset B with nine feature spaces. It was found that the proposed algorithm has the lowest root mean square error, i.e., 33% less mis-localized mobile objects. This indicates that the proposed algorithm has succeeded in improving the positioning accuracy by transferring knowledge from related source spaces to the target area. However, there is no clear evidence against the fact that negative knowledge was not transferred from the source to the target domain. In this case, the measurements that came from all Wi-Fi access points were not refined or reviewed to determine whether or not they should be used for location analysis, because they could negatively affect the positioning process. The Gaussian mixture model (GMM), however, was found to have the lowest accuracy for indoor fingerprint-based location. In about 86% of the cases, mobile devices are incorrectly localized and high computational costs are incurred. Moreover, the comparison of the accuracy of indoor positioning classifiers applied to dataset B have also been evaluated with six feature spaces. It was found that the proposed algorithm has the lowest mean square error, i.e., 22% fewer mis-localizations for mobile objects. This indicates that the proposed algorithm has succeeded in improving the positioning accuracy by transferring knowledge from related source domains to the target domain. It can be observed that not only a positive knowledge was transferred from the source domain to the prediction task, but also the computational cost and memory usage were effectively improved by excluding irrelevant features from the analysis. Similarly, the Gaussian mixture model (GMM) was found to have the lowest accuracy for indoor fingerprint-based location. In about 96% of the cases, it results in the incorrect positioning of mobile devices and high computational complexity.

Figure 23 illustrates the effects of different dimensions of feature spaces used for training indoor positioning classifiers, using dataset B with a dimension of 9 and 6, respectively. It was found that the proposed algorithm has the lowest root mean square error, i.e., 1.25 m and 1.42 m less mis-localizations of mobile devices for both scenarios. This indicates that the proposed algorithm has succeeded in improving the location accuracy by transferring knowledge from the source domain to the target domain, where the difference in location accuracy between the two scenarios is about 11%. This difference leads us to an important finding, namely that all classifiers achieved lower RMSE both before and after transfer learning for the new feature spaces of dimension 9. However, there is no strong evidence that negative knowledge was not transferred from the source to the target domain. In this case, especially for the scenario of dataset A with nine feature spaces, the measurements coming from all Wi-Fi access points were not refined or reviewed to determine whether or not they should be used for location analysis because they could negatively affect the positioning process. Moreover, the contributions of these features to target detection were not as significant (since their co-occurrence weight scores of those three features were almost negligible) as for the feature spaces with a dimension of 6. This could justify that a model with a lower RMSE does not mean that it is always the most consistent model, but that we need to examine the parameters for their negative effects on target prediction.

Figure 24a,b and Figure 25a,b illustrate the use of box and whisker (commonly known as a box plot) to compare the distributions of the RSS measurements received from an access point in a reference point of both training and testing instances, respectively. For both the training and testing instances, we have considered the measurements received from Wi-Fi AP_1 and Wi-Fi AP_48 to compare the nature of the signal distributions in terms of the box-plot parameters. One can observe that the signal distributions of reference point 1 for Wi-Fi AP_1 seem to have outlier values (both types of outliers: low and high), because the RSS values lies out of the whisker box such that the RSS measurements either fall more than 1.5 times the interquartile range above the third quartile or fall more than 1.5 times the interquartile range below the first quartile. Inter quartile range is the difference between the third and first quartile, which used to give the dispersion of a distribution. On the other side, Targets of 185, 191 and 210 have a lower outlier. One can also see that the respective medians of each box plot of a target lies outside of any box of a comparison boxplot of any target. Thus, the unique pattern for the measurements collected by the Wi-Fi APs seems to have different median values for the ‘signal-to-location’ relationships of the different target points. In other words, the received signals for a reference point from multiple APs are likely to be independent and it is what is really required in practice for reducing extra Wi-Fi Aps’ deployment. Similarly, one can compare the distribution nature in terms of skewness (positive and negative), symmetry, potential outliers, and dispersion of measurements. For instance, the measurements for targets 5 and 10 of Wi-Fi AP_4 of both training and testing instances revealed that the destruction is highly dispersed and both distributions are negatively skewed. Moreover, the measurements’ distribution received from Wi-Fi access points in both phases of instances are highly dispersed as the whisker box plot is longer. This also justifies the fact that signals were not received from this access point for their corresponding targets with the exception of a few measurements, while this automatically inflates the deviation and degrades the positioning performance.

### 4.4. Analysis of Computational Complexity 

Figure 26 presents the computational complexity of the proposed algorithm for IPS-based RSS fingerprinting, which fuses source and target domain knowledge to enhance positioning performance in the target domain. The algorithms presented in this paper were run on a laptop with an AMD Ryzen 3 3200U CPU running at 2.60 GHz and 16 GB of RAM. Two situations of complexity—time and space—are the main determinants of an algorithm’s complexity. To assess the time complexity of the algorithms, we employed both the functional analysis of Big O and the typical execution testing time. The overall worst-case computational cost analysis is given by the Big O functional analysis of time complexity, and all of the algorithms we used have an average of worst-case time complexity of $O(n^{2}).$ However, compared to the other algorithms we used, the proposed algorithm’s average elapsed execution testing time is substantially longer. Additionally, employing the two real-world dataset scenarios, Figure 26 shows the comparative study of computational testing time for FBIPS. Thus, the computational testing time for the proposed algorithm is higher, followed by the GMM, MLP, and the SVC.

## 5. Conclusions

In this paper, we propose OHetTLAL, a novel technique for indoor fingerprint-based positioning problems that can improve learning performance in the target domain through knowledge transfer from the source domain. To this end, we derived new feature spaces (on which the model is trained) based on the co-occurrence of RSS measurements from mobile devices in the two domains to capitalize on knowledge that could improve target location prediction. To derive the new feature spaces, the co-occurrence measure of the features ($C_{mi}^{p}$) was computed. The higher the value of co-occurrence between the two domains, the more detected Wi-Fi APs are shared by these two domains and the more likely these source domains are related to the target domain, which positively affects the prediction of the target. The proposed objective function was minimized across the new feature spaces so that the positive knowledge of the related source domain received a higher weight and is transferred to the target domain for predicting a new mobile device. Experimental results demonstrate that the proposed algorithm was the best algorithm for fingerprint-based indoor positioning with the minimum root mean square error in all cases of the two real office scenarios. On the other hand, the Gaussian mixture model (GMM), had the highest mis-localization error rate. Our research shows that in OHetTL, the construction of new feature spaces is critical for extracting relevant information for effective target location prediction. It should be noted that these experimental results may differ in other test phase environments. 

The effects of different dimensions of feature spaces used for training indoor positioning classifiers have also been evaluated using two real life datasets. Results demonstrate that the impact of different feature spaces used to train indoor positioning classifiers using dataset A with dimensions of 41 and 15, respectively. It was found that the proposed algorithm has the lowest root mean square error, i.e., 32% and 25% less error in positioning the mobile device for both scenarios. Similarly, the effects of different dimensions of feature spaces are used for training indoor positioning classifiers, using dataset B with a dimension of 9 and 6, respectively. It was found that the proposed algorithm has the lowest root mean square error, i.e., 33% and 22% less mis-localizations of mobile devices for both scenarios. This indicates that the proposed algorithm has succeeded in improving the positioning accuracy by transferring knowledge from the source domain to the target domain for both datasets after new feature spaces are derived based on the weight co-occurrence of both instances. It can be observed that not only positive knowledge from the source domain was transferred to the prediction task, but also the computational cost and memory consumption were effectively improved by excluding irrelevant features from the analysis. The difference in positioning accuracy obtained due to the feature spaces between the two scenarios is about 7% and 11% for both datasets of A and B, respectively. This difference leads us to an important finding, namely that all classifiers achieved lower RMSE both before and after transfer learning over the new feature spaces of dimension 9. However, there is no strong evidence that negative knowledge was not transferred from the source to the target domain. In this case, especially for the scenario of dataset B with nine feature spaces, the measurements coming from all Wi-Fi access points were not refined or scrutinized to determine whether or not they should be used for location analysis because they could negatively affect the positioning process. However, the contributions of these features to target detection were not as significant (since their co-occurrence weight scores of those three features were almost negligible) as for the feature spaces with a dimension of 6. This could justify that a model with a lower RMSE does not mean that it is always the most consistent model, but that we need to examine the parameters for their negative effects on target prediction. Our results demonstrate, however, that the proposed algorithm is more resilient to fluctuating environments and mitigates the problem of overfitting the model. Furthermore, the OHetTLAL-based fusion of both knowledge domains could significantly improve the positioning performance without any extra cost associated to offline calibrations effort. It is not necessarily the positioning performance to improve as the number of feature spaces increases, instead, how the features are important in capturing the maximum variation of the model is the main determinant.

In summary, the positioning performance, as per our results, revealed that it is highly important to extract the most significant predictors that could capture the maximum variance of the model using some metrics in order to evaluate the significance of each Wi-Fi AP deployment. Thus, we discovered that not all available feature spaces need be considered for positioning estimation for at least four reasons: (a) some features may be irrelevant and result in fingerprint duplication or pattern fingerprint mismatch; (b) large feature spaces necessitate a massive deployment of Wi-Fi access points, which is prohibitively expensive from a cost standpoint; (c) from a technical standpoint, computational cost and memory usage are quite expensive when dealing with higher dimensions of features, and (d) model over-fitting is also a serious issue for higher dimensions of feature spaces.

## Figures and Tables

**Figure 1 sensors-22-09044-f001:**
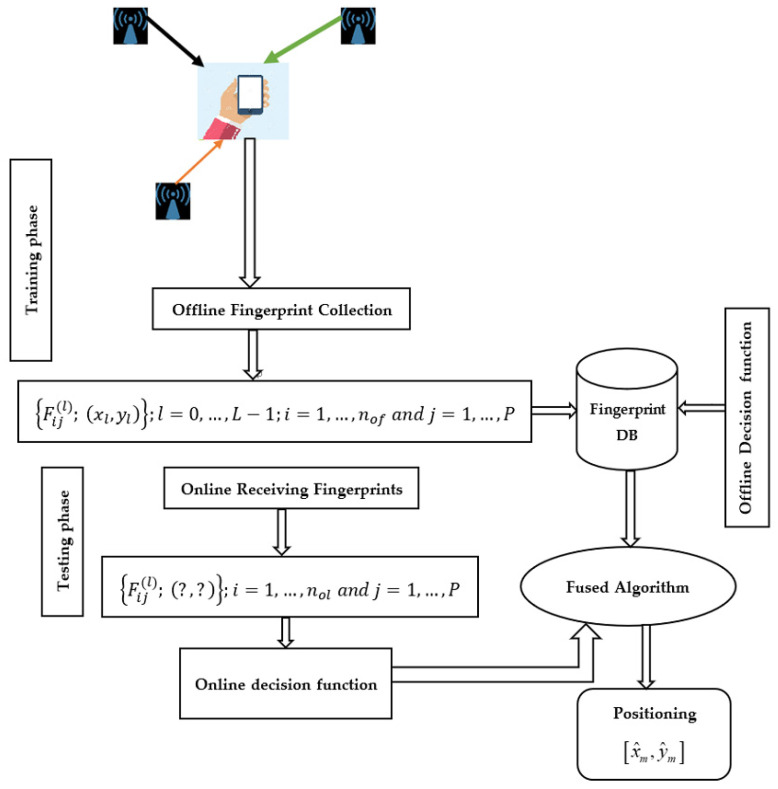
System Operation for RSS-based Fingerprinting of Indoor Positioning System.

**Figure 2 sensors-22-09044-f002:**
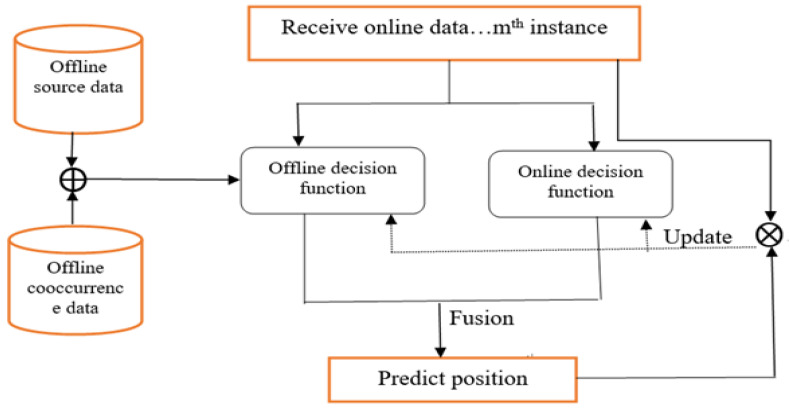
Proposed framework of heterogenous transfer learning-based weight co-occurrence for indoor positioning.

**Figure 3 sensors-22-09044-f003:**
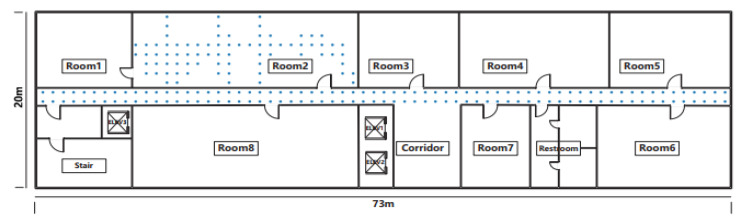
Experimental environments for dataset A showing the RP locations in blue circles.

**Figure 4 sensors-22-09044-f004:**
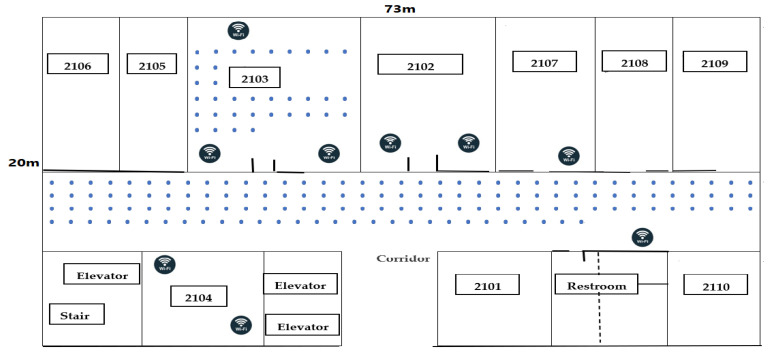
Experimental layout conducted in UESTC for generating dataset B.

**Figure 5 sensors-22-09044-f005:**
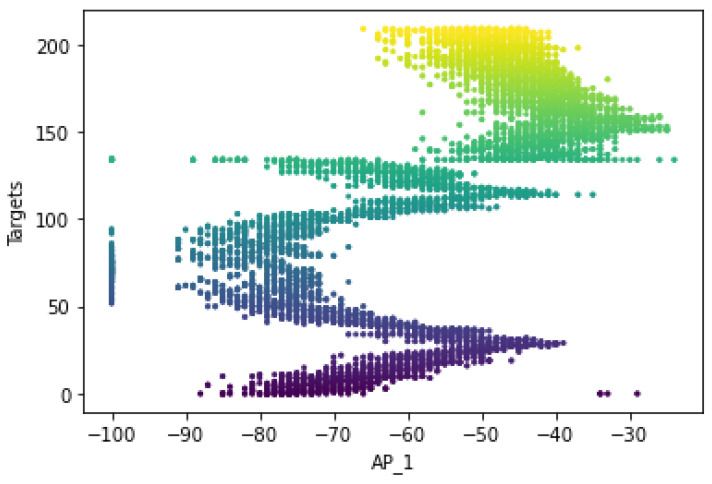
Distributions of RSS for Wi-Fi AP1 with their corresponding targets for dataset A.

**Figure 6 sensors-22-09044-f006:**
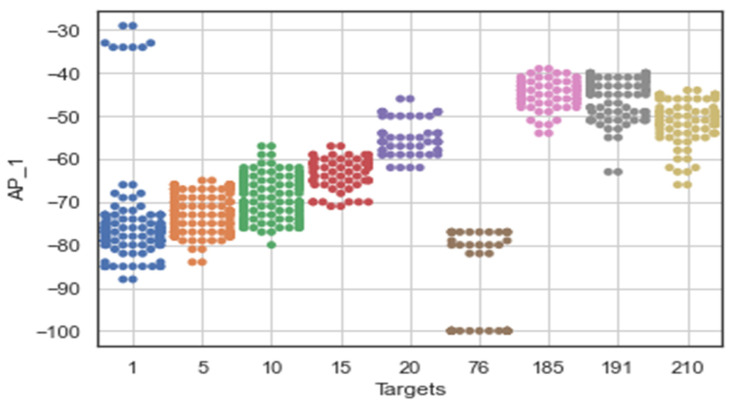
Distributions of RSS for Wi-Fi AP1 with some portion of the targets for dataset A.

**Figure 7 sensors-22-09044-f007:**
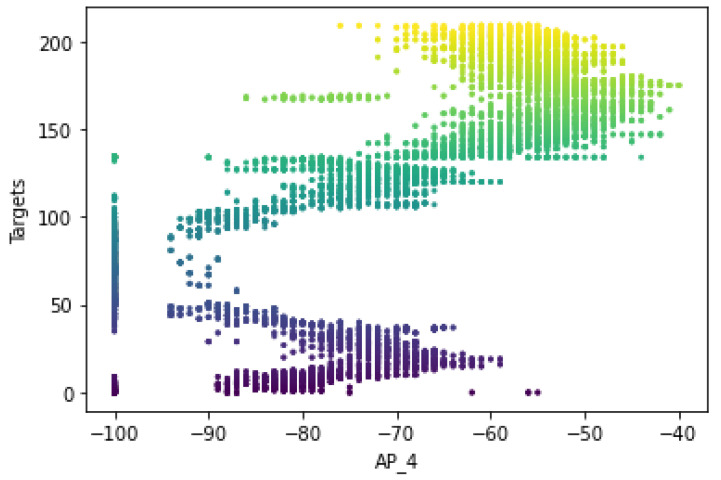
Distributions of RSS for Wi-Fi AP4 with their corresponding targets for dataset A.

**Figure 8 sensors-22-09044-f008:**
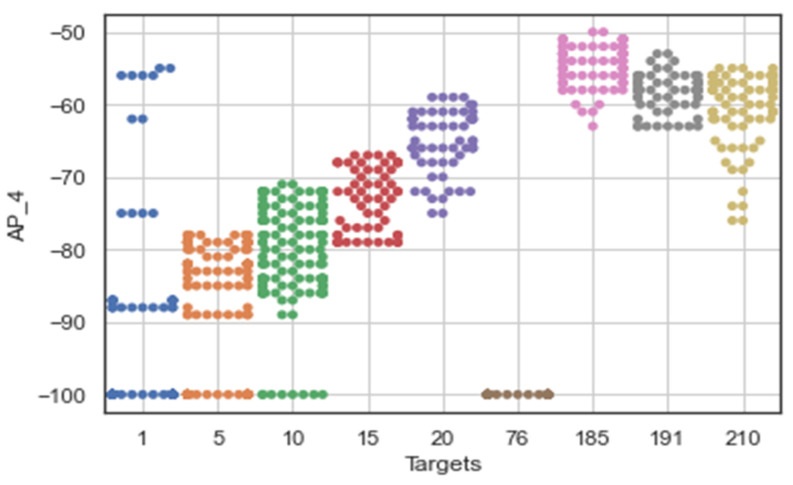
Distributions of RSS from Wi-Fi AP4 for some portions of the targets of dataset A.

**Figure 9 sensors-22-09044-f009:**
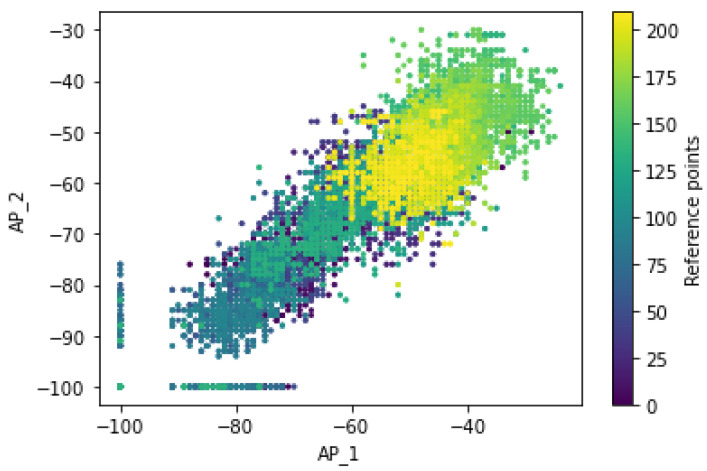
Distributions of RSS for Pair Wi-Fi APs of 1 and 2 for dataset A.

**Figure 10 sensors-22-09044-f010:**
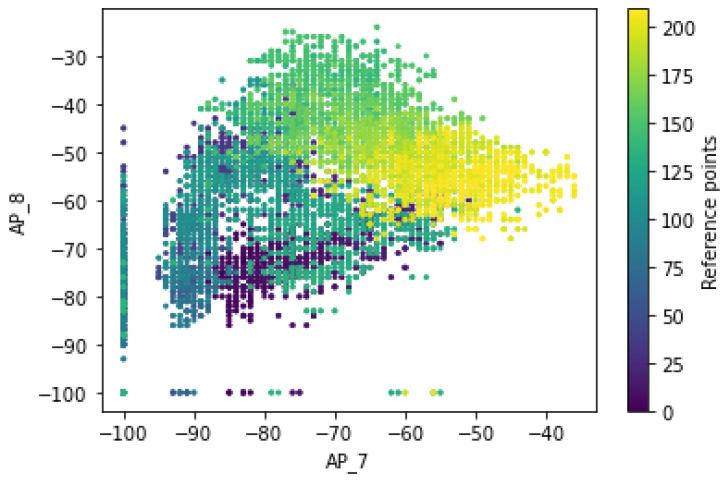
Distributions of RSS for Pair Wi-Fi APs of 7 and 8 for dataset A.

**Figure 11 sensors-22-09044-f011:**
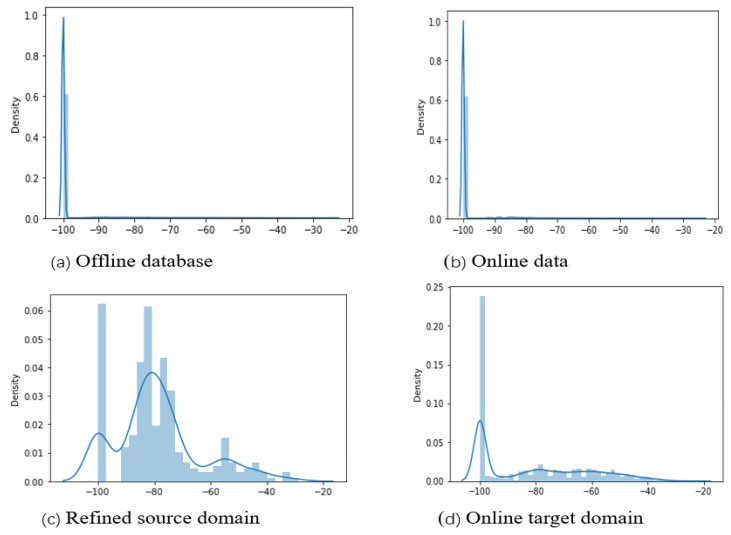
Distributions of RSS Wi-Fi APs of offline and online databases before and after transformation.

**Figure 12 sensors-22-09044-f012:**
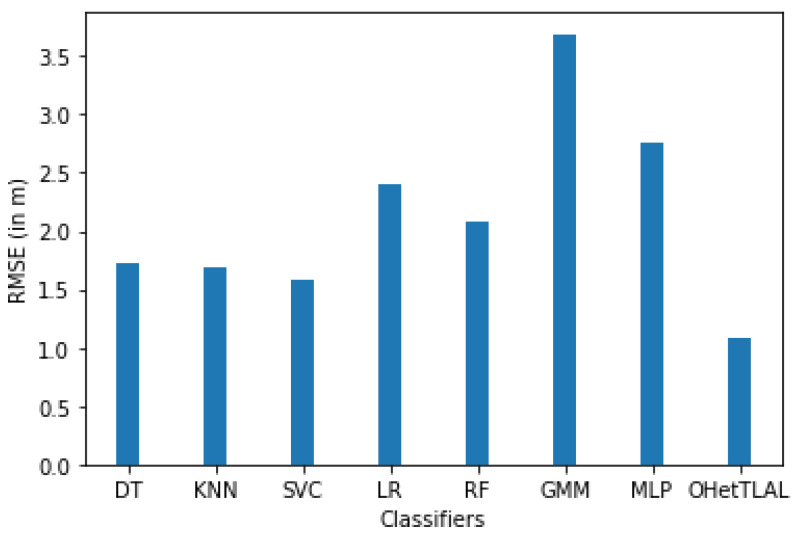
Comparisons of classifier’s RMSE for dataset A with 41 Wi-Fi APs.

**Figure 13 sensors-22-09044-f013:**
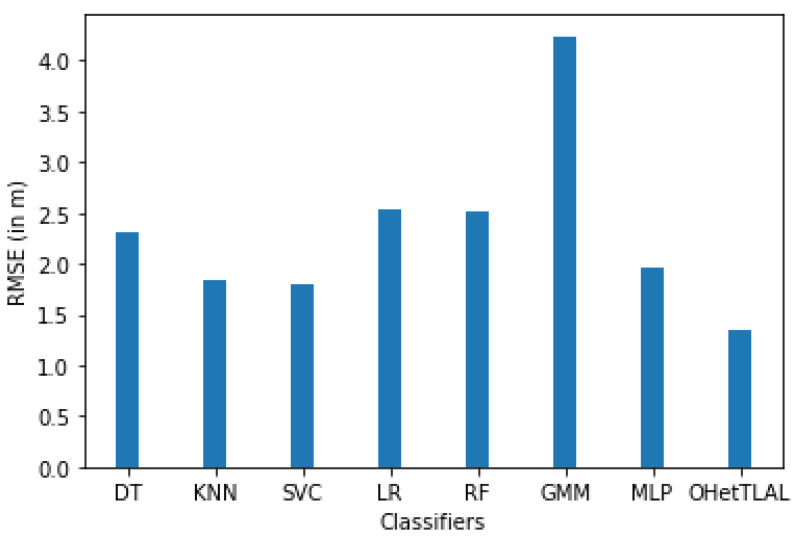
Comparisons of classifier’s RMSE for dataset A with 15 Wi-Fi APs.

**Figure 14 sensors-22-09044-f014:**
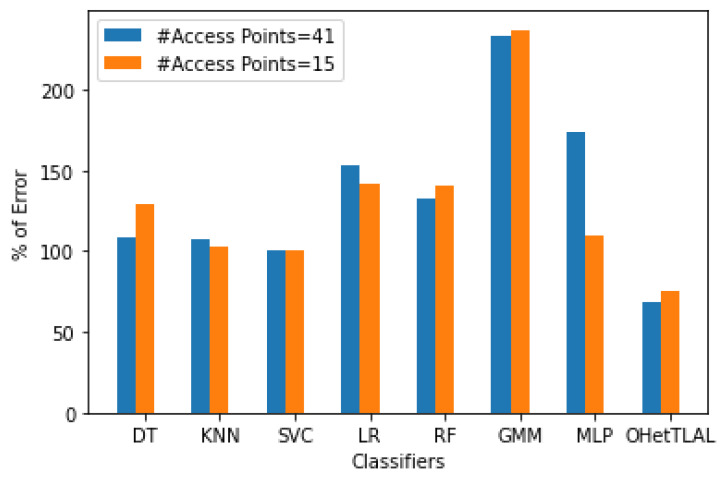
Comparison of performance of the classifiers for dataset A with 41 and 15 APs.

**Figure 15 sensors-22-09044-f015:**
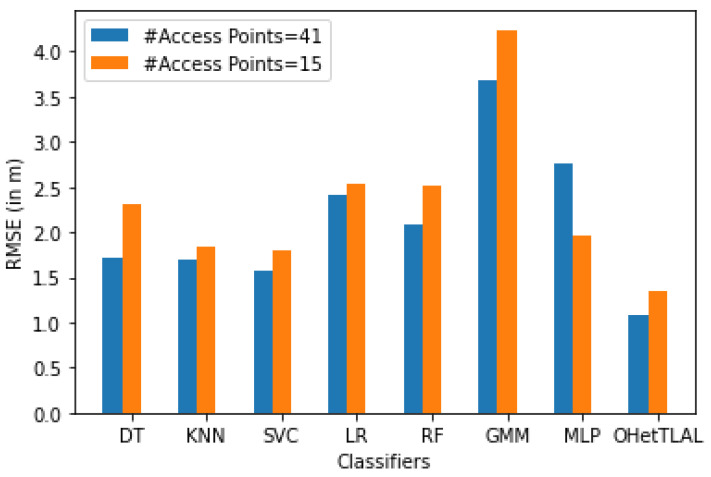
Effect of different feature spaces used to train classifiers for dataset A.

**Figure 16 sensors-22-09044-f016:**
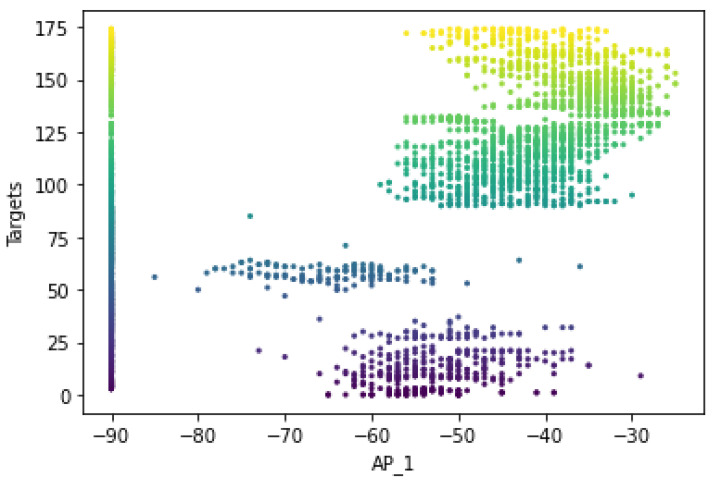
Distributions of RSS for Wi-Fi-AP1 with their corresponding Targets for dataset B.

**Figure 17 sensors-22-09044-f017:**
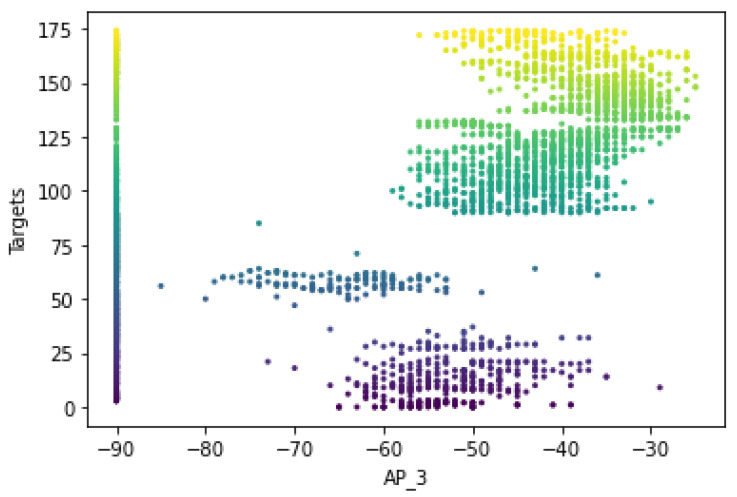
Distributions of RSS for Wi-Fi-AP3 with their corresponding targets for dataset B.

**Figure 18 sensors-22-09044-f018:**
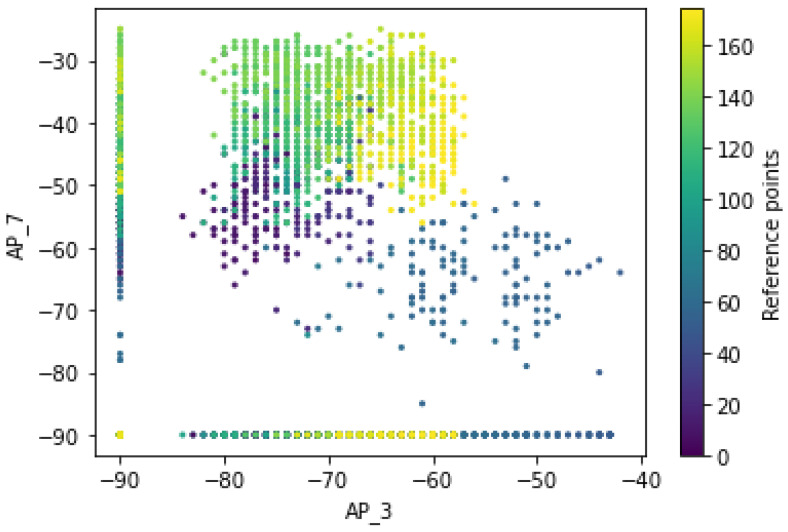
Distributions of RSS for Pair Wi-Fi AP_3 and AP_7 for dataset B.

**Figure 19 sensors-22-09044-f019:**
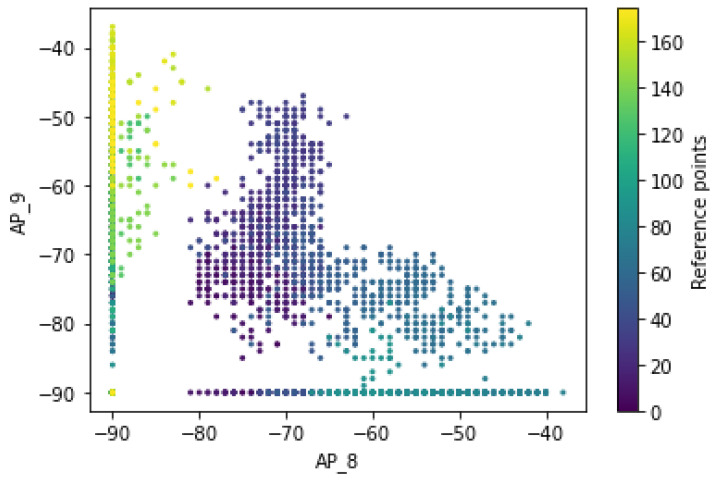
Distributions of RSS for Pair Wi-Fi APs of AP_8 and AP_9 for dataset_B.

**Figure 20 sensors-22-09044-f020:**
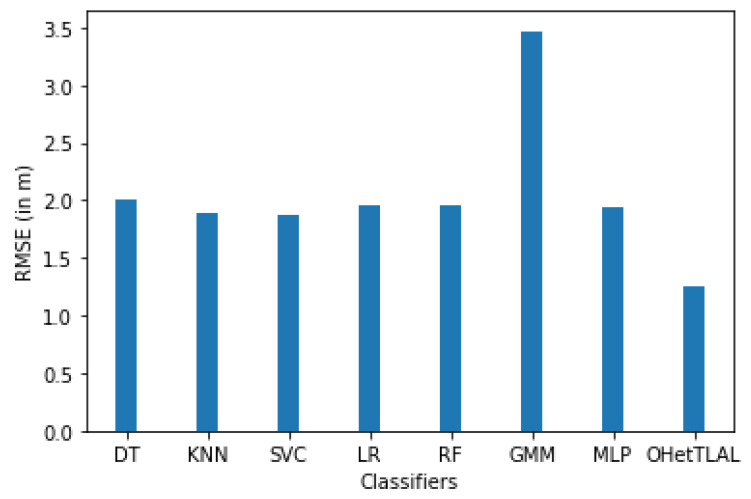
Comparisons of classifier’s RMSE (in meter) for dataset B with 9 Wi-Fi APs.

**Figure 21 sensors-22-09044-f021:**
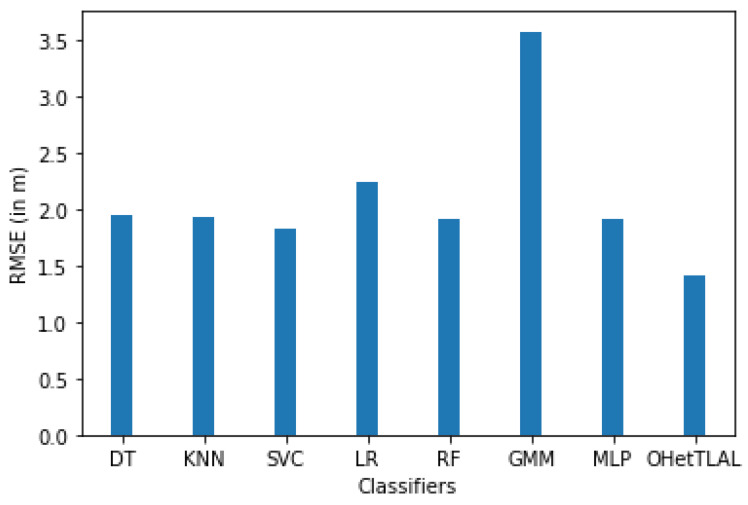
Comparisons of classifier’s RMSE (in meter) for dataset B with 6 Wi-Fi APs.

**Figure 22 sensors-22-09044-f022:**
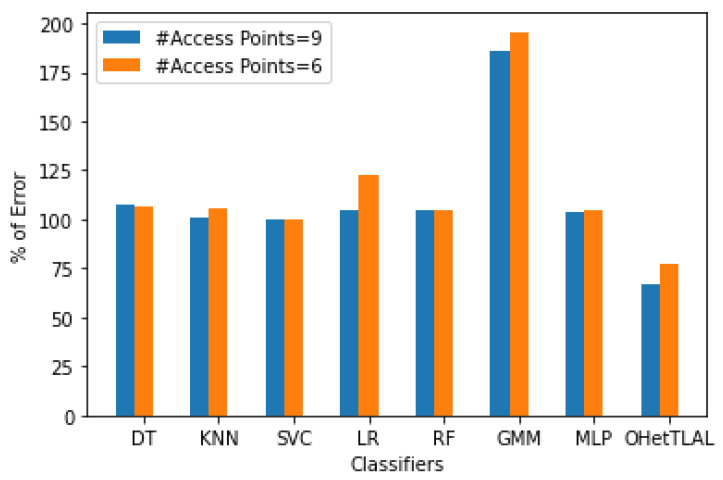
Comparison of performance of the classifiers for dataset B with 9 and 6 APs.

**Figure 23 sensors-22-09044-f023:**
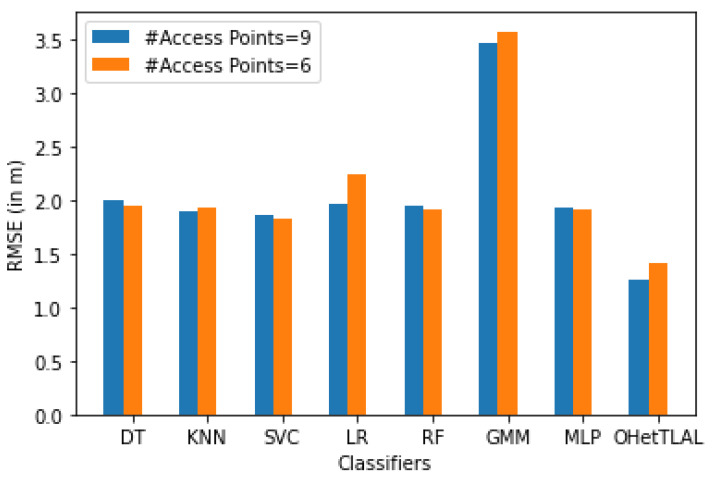
Effect of different feature spaces used to train classifiers for dataset B.

**Figure 24 sensors-22-09044-f024:**
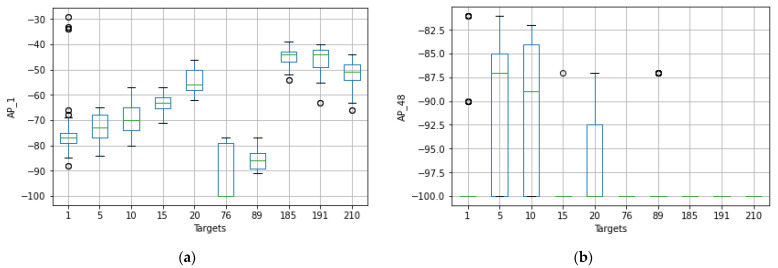
Distribution analysis of Wi-Fi RSS training measurements of different APs based on box and whisker plot. The small circles found above and below the whisker box plot represents the outliers of the training measurements received from the Wi-Fi APs of (**a**) AP_1 and (**b**) AP_48.

**Figure 25 sensors-22-09044-f025:**
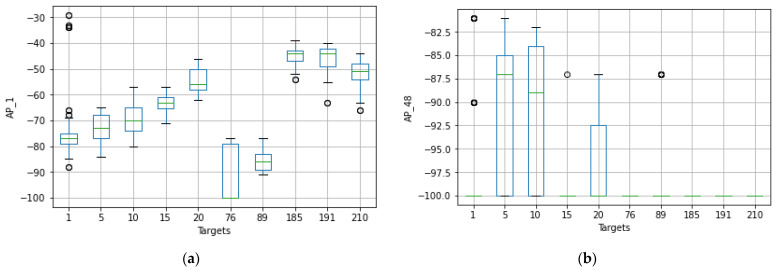
Distribution analysis of Wi-Fi RSS testing measurements of different APs based on box and whisker plot. The small circles found above and below the whisker box plot represents the outliers of the testing measurements received from the Wi-Fi APs of (**a**) AP_1 and (**b**) AP_48.

**Figure 26 sensors-22-09044-f026:**
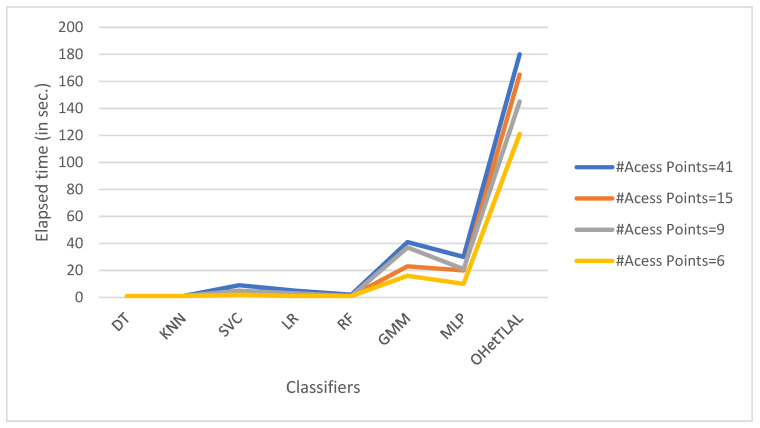
Comparative analysis of computational testing time of the algorithms used for FBIPS.

**Table 1 sensors-22-09044-t001:** Descriptive of the two real-office experiments at UESTC.

Dataset	Metrics	Area (m^2^)	UE	Base Station (BS)	#BSs	#RPs
A	RSS	1460	Smartphone	APs	416	210
B	RSS	1460	Smartphone	APs	9	175

UE: User Equipment.

**Table 2 sensors-22-09044-t002:** The distribution and descriptive measures of the Offline RSS measures.

SNo.	AP_0	AP_1	AP_2	AP_3	AP_4	AP_412	AP_413	AP_414	AP_415
Count	15,660	15,660	15,660	15,660	15,660	15,660	15,660	15,660	15,660
Mean	−59.4	−64.9	−70.1	−74.2	−73.3	−99.9	−99.9	−99.9	−99.9
std	16.8	15.9	14.5	17.9	17.1	0.11	0.43	0.15	0.27
Min	−100.0	−100.0	−100.0	−100.0	−100.0	−100.0	−100.0	−100.0	−100.0
25%	−73.0	−76.0	−81.0	−90.0	−86.0	−100.0	−100.0	−100.0	−100.0
50%	−57.0	−63.0	−71.0	−73.0	−70.0	−100.0	−100.0	−100.0	−100.0
75%	−45.0	−52.0	−58.0	−58.0	−59.0	−100.0	−100.0	−100.0	−100.0
Max	−24.0	−30.0	−36.0	−40.0	−36.0	−92.0	−93.0	−94.0	−92.0

**Table 3 sensors-22-09044-t003:** The distribution and descriptive measures of the Online RSS measures.

SNo.	AP_0	AP_1	AP_2	AP_3	AP_4	AP_412	AP_413	AP_414	AP_415
Count	4338	4338	4338	4338	4338	4338	4338	4338	4338
Mean	−58.2	−64.37	−76.53	−73.52	−100.0	−100.0	−100.0	−100.0	−100.0
std	15.63	15.14	14.40	17.92	0.00	0.00	0.00	0.00	0.00
min	−88.0	−100.0	−100.0	−100.0	−100.0	−100.0	−100.0	−100.0	−100.0
25%	−73.0	−78.0	−89.0	−88.0	−100.0	−100.0	−100.0	−100.0	−100.0
50%	−56.0	−63.0	−77.0	−72.0	−100.0	−100.0	−100.0	−100.0	−100.0
75%	−45.0	−52.0	−65.0	−58.0	−100.0	−100.0	−100.0	−100.0	−100.0
max	−24.0	−32.0	−38.0	−40.0	−100.0	−100.0	−100.0	−100.0	−100.0

**Table 4 sensors-22-09044-t004:** The distribution and descriptive measures of the Offline RSS measures.

SNo.	AP_0	AP_1	AP_2	AP_3	AP_4	AP_38	AP_39	AP_40	AP_41
Count	15,660	15,660	15,660	15,660	15,660	15,660	15,660	15,660	15,660
Mean	−59.3	−64.9	−70.1	−74.2	−81.1	−98.9	−98.9	−95.32	−94.2
std	16.7	15.8	14.5	17.9	14.5	3.07	4.07	8.07	6.77
Min	−100.0	−100.0	−100.0	−100.0	−100.0	−100.0	−100.0	−100.0	−100.0
25%	−73.0	−76.0	−81.0	−90.0	−100.0	−100.0	−100.0	−100.0	−100.0
50%	−57.0	−63.0	−71.0	−73.0	−81.00	−100.0	−100.0	−100.0	−100.0
75%	−45.0	−52.0	−58.0	−58.0	−68.0	−100.0	−100.0	−91.0	−89.0
Max	−24.0	−30.0	−36.0	−40.0	−50.0	−83.0	−68.0	−66.0	−69.0

**Table 5 sensors-22-09044-t005:** The distribution and descriptive measures of the Online RSS measures.

SNo.	AP_0	AP_1	AP_2	AP_3	AP_4	AP_38	AP_39	AP_40	AP_41
Count	4338	4338	4338	4338	4338	4338	4338	4338	4338
Mean	−58.2	−64.3	−76.5	−73.5	−80.40	−98.9	−98.3	−91.4	−93.8
std	15.6	15.14	14.4	17.9	16.3	3.7	4.7	9.0	8.1
Min	−88.0	−100.0	−100.0	−100.0	−100.0	−100.0	−100.0	−100.0	−100.0
25%	−73.0	−78.0	−89.0	−88.0	−100.0	−100.0	−100.0	−100.0	−100.0
50%	−56.0	−63.0	−77.0	−72.0	−82.00	−100.0	−100.0	−91.0	−100.0
75%	−45.0	−52.0	−65.0	−58.0	−65.0	−100.0	−100.0	−84.0	−86.0
Max	−24.0	−32.0	−38.0	−40.0	−45.0	−82.0	−79.0	−69.0	−69.0

**Table 6 sensors-22-09044-t006:** Distributions of Wi-Fi received signal strength’s per label.

Dataset A (#Labels = 210)	Labels	134	210	127	-	--	51	59	101
	# of RSS values per label	342	239	218			60	60	60
Dataset B (#Labels = 175)	Labels	134	174	127	-	--	51	59	101
	# of RSS values per label	30	30	30			30	30	30

**Table 7 sensors-22-09044-t007:** Models’ target prediction using the new feature spaces of 41 Wi-Fi access points based OHetTLAL.

Classifiers	RMSE (in Meters)
Decision tree	1.72
K -Neighbour (KNN)	1.69
Support vector machine (SVC)	1.58
Logistic Regression (LR)	2.41
Random forest	2.09
Gaussian Mixture	3.69
Neural Network (MLP)	2.75
The proposed algorithm (based TL)	1.08

Transfer Learning (TL) based.

**Table 8 sensors-22-09044-t008:** Models’ target prediction using the new feature spaces of 15 Wi-Fi access points based on OHetTLAL.

Classifiers	RMSE (in Meters)
Decision tree	2.31
K -Neighbour (KNN)	1.83
Support vector machine (SVC)	1.79
Logistic Regression (LR)	2.53
Random forest	2.52
Gaussian Mixture	4.24
Neural Network (MLP)	1.97
The proposed algorithm (based TL)	1.35

**Table 9 sensors-22-09044-t009:** Models’ target prediction using the original feature spaces of 9 Wi-Fi access points based on OHetTLAL.

Classifiers	RMSE (in Meters)
Decision tree	2.01
K -Neighbour (KNN)	1.89
Support vector machine (SVC)	1.87
Logistic Regression (LR)	1.96
Random forest	1.95
Gaussian Mixture	3.47
Neural Network (MLP)	1.94
The proposed algorithm (based TL)	1.25

**Table 10 sensors-22-09044-t010:** Models’ target prediction using the new feature spaces of 6 Wi-Fi access points based on OHetTLAL.

Classifiers	RMSE (in Meters)
Decision tree	1.95
K -Neighbour (KNN)	1.93
Support vector machine (SVC)	1.83
Logistic Regression (LR)	2.24
Random forest	1.92
Gaussian Mixture	3.58
Neural Network (MLP)	1.91
The proposed algorithm (based TL)	1.42

## Data Availability

The dataset used for this study are available upon request to the corresponding author.

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
