# Peer review of "OHetTLAL: An Online Transfer Learning Method for Fingerprint-Based Indoor Positioning"

_sensors, 2022, doi:10.3390/s22239044_

Round 1

Reviewer 1 Report

The authors proposed an online heterogeneous transfer learning (OHetTLAL) algorithm for IPS -based RSS fingerprinting to improve the positioning performance in the target domain by fusing both source and target domain knowledge. The paper seems well organized and has good technical sounds, but I have some comments and suggestions:

- All the figures in section 4 are very low quality and not acceptable. 

- The RMSE of the proposed algorithm is more than 1 meter which considers a very high localization error since you used 9 APs. Please see other related works such as "Robust 3D indoor positioning system based on radio map using Bayesian network" and "Improving accuracy in indoor localization system using fingerprinting technique"

- The authors should provide any results to show the effect of increasing /decreasing APs and  RPs on the system accuracy. 

- I do not see any of the overall system accuracies of using two different datasets. 

- How about comparing your proposed algorithm with the CNN model? 

Author Response

Reviewer #1:

Notes on Revision

Re: Manuscript sensors-194-8894, entitled OHetTLAL: An Online Transfer Learning method for Fingerprint-based Indoor Positioning

We would like to express our gratitude to the editors and the anonymous reviewers for their constructive suggestions and criticism. The comments are well taken, and the manuscript has been revised accordingly. Below please find our responses to the reviewers’ comments. Also, for the reviewers’ convenience, major changes are written in YELLOW in the revised manuscript.

Responses to the Comments of Reviewer 1

We thank the Reviewer for the comments, suggestions and questions that helped us to improve the quality of the manuscript.

Comments and Suggestions for Authors

The authors proposed an online heterogeneous transfer learning (OHetTLAL) algorithm for IPS -based RSS fingerprinting to improve the positioning performance in the target domain by fusing both source and target domain knowledge. The paper seems well organized and has good technical sounds, but I have some comments and suggestions:

Comment #1: All the figures in section 4 are very low quality and not acceptable. 

Response #1: Thank you for your suggestions. In the revised version, we have corrected all the Figures.

Comment #2:  The RMSE of the proposed algorithm is more than 1 meter which considers a very high localization error since you used 9 APs. Please see other related works such as "Robust 3D indoor positioning system based on radio map using Bayesian network" and "Improving accuracy in indoor localization system using fingerprinting technique"

Response #2: Thank you for the important references that helped us to gain insight in our work. In the revised version, we have used them to describe more about the works related to our focus.

Comment #3: The authors should provide any results to show the effect of increasing /decreasing APs and  RPs on the system accuracy. 

Response #3: Thank you for your suggestions. In the revised version, we have used a Figures 15, 23 that compares the effect of the number of access points in relation to their performance.

Comment #4: I do not see any of the overall system accuracies of using two different datasets

Response #4: Thank you for your suggestion. We used the two datasets to demonstrate the possibility of knowledge transfer to target domain from two experiment settings. For validation purpose. This is why we considered various scenarios to learn the predictive model using different measurements to validate our results.

Comment #5: How about comparing your proposed algorithm with the CNN model?

Response #5: Thank you for your suggestion. We also share this idea of using CNN model as a future work.

Reviewer 2 Report

The article presents an approach to indoor localization based on signal strength measurements, fingerprinting, and transfer learning. The idea is quite clearly presented but the proposed approach lacks details and clarity. Authors need to improve the article before it is resubmitted.

Evaluation scenarios are not clearly presented:

1. what number of APs and how are they deployed in the evaluation areas A and B?  Do you have 416 APs in approx 1400 square meter area?

2. for test scenario B only 9 APs were used, but the test area (and test points) were the same as for scenario A? How did you choose the 9 APs used? Are these 9 APs a subset of 41 or 15 APs used in scenario A?

3. in the evaluation APs and mobile phones are used but it is not clear how the system operates - which devices measure the RSS (APs or mobile)? Does the system work by transmitting signals from APs to devices, or from devices to APs. Please explain the operation of the system.

4. article mentions both APs and "base stations" - what are the base stations, and how they are different from APs and mobile devices? E.g. there is a sentence saying:  "... the base stations are not detected at all for the majority of the Wi-Fi APs."

The proposed approach requires clarification:

5. do I understand correctly that the feature selection is actually about deciding which APs, out of all available APs, to choose to get the best localization accuracy? The article is missing a clear, high-level description of the proposed approach. 

6. if the above understanding is correct then it would be interesting to see which 15 APs out of 41 and 416 were chosen and give better accuracy. This might give some practical information on how APs should be deployed in the area to optimize for accuracy. 

7. you choose 15 out of 41 and 41 out of 416 features to use based on co-occurrence analysis. How did you come up with the threshold for the co-occurrence or how did you decide the number of features to be used (why 15, would 14 or 16 features give a better result)?

8. some of the symbols used in the formulas are not explained in the text - please make sure that the meaning of all the symbols is clear to the reader

9.  is the RSS value of -100 a substitute for missing measurement (the device has not received a signal from the AP and you have assumed the value of RSS is equal to -100) or it is a value measured and reported by the device?

10. in both cases above the standard deviation of such RSS will be small - the device is either at the edge of the communication range or is not heard at all and fake values are set. Consequently, I don't agree with the statement: "The lowest values of standard deviation observed in the above scenario suggest a higher consistency of measurements".

11. Why did you decided to use RSS not the CSI (channel state information) measurements? According to literature CSI is more stable and allows to achieve higher accuracies.

Analysis of the datasets

12. the analysis of the data sets is definitely too long and contains repetitions - what is different about figs 4, and 5 and the corresponding description in the text, that justifies the need to include both figures in the article? It seems the conclusions of these paragraphs are the same. Similar is the case for figs 6 and 7.

13. Fig 8a and the corresponding description in the text suggest that the vast majority of 416 APs have measured RSS values close to -100. How this is possible? How this relates to the previous analysis in the article and plots 4-7?

Presentation of the results:

14. the analysis only presents the RMSE while in the domain of localization articles often report median, and percentiles of the error (in particular 90th percentile is of interest) as this gives a better overview and understanding of what can be accepted in the application. I would appreciate a more detailed analysis of the localization accuracy

15. classifiers used for comparison/evaluation - were they implemented by the authors or they are taken from the literature? How does the proposed algorithm compare to other methods presented in the literature? One option would be to use publicly available RSS measurements for some scenarios and compare results reported in the literature with your approach. Without this reader cannot tell how good is the proposed algorithm overall.

16. the need for plots 23 and 24 is not clear - please consider if they are needed

The article requires significant editorial work:

17. figures are generally of very low quality and need to be improved - please increase  the resolution (consider the use of vector graphics), correct readability of the text on figures, and scale figures properly. Text on the figures should be the same size (at least similar) as in the article.

18. Plots 10-13 should not have lines that connect error values for different methods. Lines do not make sense for these plots, you can use bars instead. The same for figures 18-22

19. please make sure all axis on the figures have a description of the axis and units.

20. please use \begin{equation} for all numbered equations, and $ signs for in-line equations. Do not use images. Make sure symbols used in equations are clearly explained in the text and are consistent across the article

21. I would recommend extending the captions of the figures and tables to be more informative.

Author Response

Reviewer #2

Notes on Revision

Re: Manuscript sensors-194-8894, entitled OHetTLAL: An Online Transfer Learning method for Fingerprint-based Indoor Positioning

We would like to express our gratitude to the editors and the anonymous reviewers for their constructive suggestions and criticism. The comments are well taken, and the manuscript has been revised accordingly. Below please find our responses to the reviewers’ comments. Also, for the reviewers’ convenience, major changes are written in YELLOW in the revised manuscript.

Responses to the Comments of Reviewer 2

We thank the Reviewer for the comments that helped us to improve the quality of the manuscript.

Comments and Suggestions for Authors

The article presents an approach to indoor localization based on signal strength measurements, fingerprinting, and transfer learning. The idea is quite clearly presented but the proposed approach lacks details and clarity. Authors need to improve the article before it is resubmitted.

Evaluation scenarios are not clearly presented:

Comment #1: what number of APs and how are they deployed in the evaluation areas A and B?  Do you have 416 APs in approx 1400 square meter area? 1460m2

Response #1: Thank you for your valuable comment. Two real-world experiments were carried out to evaluate our proposed algorithm. Experimental settings and datasets were first presented, and the overall performance of the classifiers are analyzed. Both feature spaces of scenarios were considered (before and after transformation). The general layout schema of the two experiments generating Dataset A and B are described below:

Dataset A: The experiment was conducted at UESTC (University of Electronic Science and Technology of China) as depicted in Fig. 2 and the 21st floor of the Innovation building was taken as the indoor environment site with 416 APs, consisting of 10 offices and one corridor covers an area of 1460 m2 and partitioned into 210 reference points (RP) which represent for the target’s location。 Each RP is numbered with a label .  and  be the number of offline and online instances collected during the offline training phases and the online testing phase respectively. The RSS of fingerprints at the lth RP can be represented as a matrix:  Smartphones were used for data collection at the deployment area. The 416 are number of the access points (APs) where initial position of the target or position of base stations (BS) are stored in fingerprint database. This are the feature spaces that can be used to map with the online testing RSS measurements.

Fig. 2 Experimental environments for dataset A showing the RP locations in blue circles.

Dataset B: Another real-office experiment at UESTC (University of Electronic Science and Technology of China) on the 21st floor of the Innovation Building was also conducted in different environmental settings to validate our results as shown in Fig. 3. It covers an area of 1460 m2 and partitioned into 175 reference points (RP). 9 Wi-Fi access points were sparsely deployed to guarantee that at least three access points are detectable at each RP.

Fig. 3 Experimental layout conducted in UESTC for generating Dataset B.

Moreover, Table 1 below provides the summary of the datasets.

Table 1. Descriptive of the two real-office experiment at UESTC

Dataset

Metrics

Area (m2)

UE

Base station (BS)

#BSs

#RPs

A

RSS

1460

Smartphone

APs

416

210

B

RSS

1460

Smartphone

APs

9

175

Table 5 shows the number of measurements from each reference point is different and shows the distribution is unbalanced, so feature scaling was considered to avoid the effect of dominance of the larger occurrence of labels within the cluster. Otherwise, the larger features would dominate the others within the cluster. On the other side, the number of measurements collected from each RP and demonstrates that the label distribution for dataset A is balanced and no feature scaling technique is used to avoid the dominance effect of the cluster’s higher label occurrence.

Table 5. Distributions of Wi-Fi received signal strength’s per Label

Dataset A (#Labels = 210)

Labels

134

174

127

-

--

51

59

101

# of RSS values per label

342

239

218

60

60

60

Dataset B (#Labels = 175)

Labels

134

210

127

-

--

51

59

101

# of RSS values per label

30

30

30

30

30

30

Comment #2: for test scenario B only 9 APs were used, but the test area (and test points) were the same as for scenario A? How did you choose the 9 APs used? Are these 9 APs a subset of 41 or 15 APs used in scenario A?

 Response #2: Thank you for your valuable suggestion. We have described it in your Comment #1 for the details. There are two datasets A and B as shown below and they are separate datasets. During the training phase, a survey is conducted to collect RSS measurements from predetermined known locations (mapped into 2D coordinate system), and a database of fingerprint patterns indexed with their corresponding reference points (RPs) is created. Those features constituting 416 access points are considered as base stations and used to do the mapping during the testing time.

Moreover, Table 1 below provides the summary of the datasets.

Table 1. Descriptive of the two real-office experiment at UESTC

Dataset

Metrics

Area (m2)

UE

Base station (BS)

#BSs

#RPs

A

RSS

1460

Smartphone

APs

416

210

B

RSS

1460

Smartphone

APs

9

175

Comment #3: in the evaluation APs and mobile phones are used but it is not clear how the system operates - which devices measure the RSS (APs or mobile)? Does the system work by transmitting signals from APs to devices, or from devices to APs. Please explain the operation of the system.

Response #3: Thank you for your valuable suggestion. In the revised version, we have added the following ideas such that:

“The Wi-Fi APs send the signal and the mobile user receive the RSS. The localization process done at user’s side connected to a server. Moreover, Figure 1 below explicitly describes the system operation used for RSS-based Fingerprinting of Indoor Positioning System.”

Figure 1: System Operation for RSS-based Fingerprinting of Indoor Positioning System.

Comment #4: article mentions both APs and "base stations" - what are the base stations, and how they are different from APs and mobile devices? E.g. there is a sentence saying:  "... the base stations are not detected at all for the majority of the Wi-Fi APs."

 Response #4: Thank you for your valuable suggestion. In the revised version, we have rewritten as: we used the term base stations to describe the access points (APs) where initial position of the target or position of base stations (BS) are stored in fingerprint database. This are the feature spaces that can be used to map with the online testing RSS measurements. Furthermore, we have added in the revised version to describe the detailed system operation how it works as shown in Figure 1. As Fingerprint involves two parts: the features stored in the fingerprint used for mapping during online are considered as the base stations which are collected from the APs (offline stage).

The proposed approach requires clarification:

Comment #5: do I understand correctly that the feature selection is actually about deciding which APs, out of all available APs, to choose to get the best localization accuracy? The article is missing a clear, high-level description of the proposed approach.

 Response #5: Thank you for your valuable suggestion. Of course, yes. The features are the vectors created at each reference point during the measurement collection and received from the available Wi-Fi APs.

Comment #6: if the above understanding is correct then it would be interesting to see which 15 APs out of 41 and 416 were chosen and give better accuracy. This might give some practical information on how APs should be deployed in the area to optimize for accuracy.

 Response #6: Thank you for your valuable suggestion. Here below are the selected features:

For data set A:

Case 1: 41 number of features are selected.

AP1, AP2, AP3, AP4, AP6, AP7, AP8, AP9, AP11, AP12, AP21, AP22, AP24, AP25, AP26, AP28, AP29, AP30, AP32, AP33, AP34, AP35, AP36, AP40, AP43, AP45, AP48, AP50, AP56, AP60, AP61, AP65, AP66, AP67, AP70, AP98, AP99, AP126, AP157, AP175, AP197

Case 2: 15 number of features are selected.

AP1, AP2, AP3, AP4, AP6, AP7, AP8, AP9, AP11, AP12, AP22, AP24, AP25, AP26, AP48

For data set B:

AP1, AP2, AP3, AP7, AP8, AP9 are selected as best features as their co-occurrence of both instances of offline and online have scored higher weights. The threshold is between zero and one. The closer to one is the higher weight whereas the closer to zero is the least cooccurrence.

Conclusion: Those lists for both datasets are the best features stored in the fingerprint database that do the best mapping during the online localization. We share the idea that this finding could give some practical information on how APs should be deployed in the area to optimize for accuracy.

Comment #7: you choose 15 out of 41 and 41 out of 416 features to use based on co-occurrence analysis. How did you come up with the threshold for the co-occurrence or how did you decide the number of features to be used (why 15, would 14 or 16 features give a better result)?

 Response #7: Thank you for your valuable suggestion.

Suppose there are P Wi-Fi access points (Wi-Fi APs) detected in both offline and online phases. Let be represents the set of detected Wi-Fi APs for offline and online phases respectively where  A binary response of the detection indicators would take a value of 1 when the corresponding Wi-Fi APs is detected otherwise 0 for undetected, for the pth Wi-Fi access point in the ith offline instance and mth online instance, respectively. To construct a refined source domain, a metric was used to compute the co-occurrence between offline sample and online sample, defined as:

                              (2)

We can calculate the weight co-occurrence scores for all instances of both sides of the source and target domains as:  The higher would imply the more detected Wi-Fi APs are shared by these two domains, and the more likely this offline data is useful for the target prediction. The source domain has been refined based on the target domain to avoid negative knowledge transfer. The co-occurrence measures of feature spaces ( ) were used to derive the homogeneous new feature spaces and those features with higher weight scores have got selected to construct the source domain and used as an input to train the classifier as it could improve the target’s location prediction. The pseudo code to construct source domain is also provided in Algorithm 1.

Comment #8: some of the symbols used in the formulas are not explained in the text - please make sure that the meaning of all the symbols is clear to the reader

 Response #8: Thank you for your valuable suggestion. In the revised version, we have corrected it.

Comment #9:  is the RSS value of -100 a substitute for missing measurement (the device has not received a signal from the AP and you have assumed the value of RSS is equal to -100) or it is a value measured and reported by the device?

Response #9: Thank you for your valuable comment. Yes, the RSS value of -100 was used when the device has not received a signal from the Wi-Fi AP.

Comment #10: in both cases above the standard deviation of such RSS will be small - the device is either at the edge of the communication range or is not heard at all and fake values are set. Consequently, I don't agree with the statement: "The lowest values of standard deviation observed in the above scenario suggest a higher consistency of measurements".

 Response #10: Thank you for your valuable suggestion. Our idea is completely similar with your suggestion. In the revised version, we have rewritten it in a better way. Our idea is that, by definition the lesser the coefficient of variation (CV) or standard deviation in our case, implies the higher consistency of measurements. However, as you said it, the values feed with -100 gives zero standard deviation for both instances of training and testing. Thus, we set an exclusion criterion to remove them from the further analysis. This is exactly the same idea with your suggestion. This is also achieved by the co-occurrence measures of instances we used.

Comment #11: Why did you decided to use RSS not the CSI (channel state information) measurements? According to literature CSI is more stable and allows to achieve higher accuracies.

Response #11: Thank you for your valuable suggestion. In our case, we have used the RSS fingerprint as it is compatible with the already installed infrastructure and ease of implementation which doesn’t require any extra hardware device to extract Wi-Fi RSS values unlike the CSI need NIC network cards. Otherwise, now days in modern Wi-Fi networks, CSI values can easily be extracted from the Wi-Fi networks. Thus, we share the idea that, the use of CSI could bring more an efficient estimator for location estimation.

Analysis of the datasets

Comment #12: the analysis of the data sets is definitely too long and contains repetitions - what is different about figs 4, and 5 and the corresponding description in the text, that justifies the need to include both figures in the article? It seems the conclusions of these paragraphs are the same. Similar is the case for figs 6 and 7.

 Response #12: Thank you for your valuable suggestion. In the revised version, we have merged them and provides a summarized description.

Comment #13: Fig 8a and the corresponding description in the text suggest that the vast majority of 416 APs have measured RSS values close to -100. How this is possible? How this relates to the previous analysis in the article and plots 4-7?

Response #13: Thank you for your valuable comment. For instance, dataset A, has stored fingerprints collected from all available APs. This are the feature spaces that can be used to map with the online testing RSS measurements. Thus, based on the co-occurrence metrics and exclusion criteria, we have reduced the number of features by extracting the most significant features or that could improve the mapping in the target domain. As clearly can be shown in Fig 8 (c) and (d), after the source domain refinement, we managed to extract the best features with high occurrences of both instances of training and testing. Similarly, the previous analysis in the article and plots 4-7, are among the best features which are selected based on the metrics we set. It is why, we observe a different distribution. Moreover, we have stated the features that have been selected in Comment #6.

Presentation of the results:

Comment #14: the analysis only presents the RMSE while in the domain of localization articles often report median, and percentiles of the error (in particular 90th percentile is of interest) as this gives a better overview and understanding of what can be accepted in the application. I would appreciate a more detailed analysis of the localization accuracy

 Response #14: Thank you for your valuable suggestion. In the revised version, we have used percentage error for better overview as shown in Figure 14, and 22.

Comment #15: classifiers used for comparison/evaluation - were they implemented by the authors or they are taken from the literature? How does the proposed algorithm compare to other methods presented in the literature? One option would be to use publicly available RSS measurements for some scenarios and compare results reported in the literature with your approach. Without this reader cannot tell how good is the proposed algorithm overall.

 Response #15: Thank you for your valuable suggestion. Our proposed model was compared with seven machine learning algorithms (Decision Tree (DT), K-nearest neighbor (KNN), Support Vector Machine (SVC), Logistic Regression (LR), Random Forest (RF), Gaussian Mixture Model (GMM), and Neural Network (MLP)). We have applied those popular machine learning algorithms using the two real office experiments datasets for validation and compare with the proposed algorithm.

Comment #16: the need for plots 23 and 24 is not clear - please consider if they are needed

 Response #16: Thank you for your valuable suggestion. In the revised version, we have omitted them.

The article requires significant editorial work:

Comment #17: figures are generally of very low quality and need to be improved - please increase  the resolution (consider the use of vector graphics), correct readability of the text on figures, and scale figures properly. Text on the figures should be the same size (at least similar) as in the article.

 Response #17: Thank you for your valuable suggestion. In the revised version, we have all modified them with better resolution.

Comment #`8: Plots 10-13 should not have lines that connect error values for different methods. Lines do not make sense for these plots, you can use bars instead. The same for figures 18-22

 Response #18: Thank you for your valuable suggestion. In the revised version, we have used bars accordingly.

Comment #19: please make sure all axis on the figures have a description of the axis and units.

 Response #19: Thank you for your valuable suggestion. In the revised version, we have corrected it.

Comment #20: please use \begin{equation} for all numbered equations, and $ signs for in-line equations. Do not use images. Make sure symbols used in equations are clearly explained in the text and are consistent across the article

 Response #20: Thank you for your valuable suggestion. In the revised version, we have adjusted the equations using math type and described them within the text document explicitly and consistent across the article.

Comment #21: I would recommend extending the captions of the figures and tables to be more informative.

Response #21: Thank you for your valuable suggestion. In the revised version, we have fixed it.

Reviewer 3 Report

MDPI Sensors Journal (Manuscript ID: sensors- 1955524)

Comments to the Author

This paper proposes an online heterogeneous transfer learning  algorithm for IPS-based RSS fingerprinting to improve the positioning performance in the target domain. It is an important topic and the paper studies the concept clearly. However, there are several points need to be addressed to improve the quality of the manuscript.

Suggestions to improve the quality of the paper are provided below:

1)     Title of the manuscript is a little misleading. It sounds like a review paper, whereas the authors propose an approach for indoor positioning. My suggestion is to update the title as “ An online transfer learning method for fingerprint-based indoor positioning” to make it more clear.

2)     Abstract needs to be more concise and extensive details should be removed. Currently, it contains a very detailed explanation of the feature space (line 26-31) which can be removed and elaborated more in the methodology if needed.

3)     Please mention several applications of indoor localization technologies in the building domain as it is highly valuable for building community (Line 57-58). Please refer to the highlighted building applications such as include emergency management, plug load controls, and HVAC controls.

Indoor localisation for building emergency management

Filippoupolitis, A., Oliff, W., & Loukas, G. (2016, December). Bluetooth low energy based occupancy detection for emergency management. In 2016 15th international conference on ubiquitous computing and communications and 2016 International Symposium on Cyberspace and Security (IUCC-CSS) (pp. 31-38). IEEE.

Indoor localisation for smart plug load control

Tekler, Zeynep Duygu, et al. "Plug-Mate: An IoT-based occupancy-driven plug load management system in smart buildings." Building and Environment 223 (2022): 109472. 

Indoor localisation for smart HVAC controls

Balaji, B., Xu, J., Nwokafor, A., Gupta, R. and Agarwal, Y., 2013, November. Sentinel: occupancy based HVAC actuation using existing WiFi infrastructure within commercial buildings. In Proceedings of the 11th ACM Conference on Embedded Networked Sensor Systems (pp. 1-14).

4)     Under Section 2.1 Indoor Positioning Systems, the authors mentioning about the importance of Wi-Fi based fingerprinting as it is available in Wi-Fi-enabled devices, which is true and at the same time Bluetooth Low Energy (BLE)-based technologies are also available in mobile devices and became very popular over the recent years. I highly suggest authors to include reviewing several works using BLE-based localization technologies. Some of them are given below. Please review these works under this section to enrich the content.

P. C. Ng, P. Spachos and K. N. Plataniotis, "COVID-19 and Your Smartphone: BLE-Based Smart Contact Tracing," in IEEE Systems Journal, vol. 15, no. 4, pp. 5367-5378, Dec. 2021, doi: 10.1109/JSYST.2021.3055675.

Tekler, Z.D., Low, R., Gunay, B., Andersen, R.K. and Blessing, L., 2020. A scalable Bluetooth Low Energy approach to identify occupancy patterns and profiles in office spaces. Building and Environment171, p.106681.

Baronti, Paolo, et al. "Indoor bluetooth low energy dataset for localization, tracking, occupancy, and social interaction." Sensors 18.12 (2018): 4462.

Filippoupolitis, A., Oliff, W. and Loukas, G., 2016, October. Occupancy detection for building emergency management using BLE beacons. In International Symposium on Computer and Information Sciences (pp. 233-240). Springer, Cham.

Tekler ZD, Low R, Blessing L. An alternative approach to monitor occupancy using bluetooth low energy technology in an office environment. InJournal of Physics: Conference Series 2019 Nov 1 (Vol. 1343, No. 1, p. 012116). IOP Publishing.

Huang, Ke, Ke He, and Xuecheng Du. "A hybrid method to improve the BLE-based indoor positioning in a dense bluetooth environment." Sensors 19.2 (2019): 424.

5)     There are several formatting issues throughout the manuscript.

·       Section 4.1 (Line 320) and Section 4.1.1 Dataset A (Line 321) need to be bolded as they are section headers.

·       I would suggest merging Section 4.2 (Line 338), Section 4.3 (Line 357) and naming the section as “Distribution of Homogeneous Feature Spaces. Then, you can divide it into subsections such as “Before Transformation” and “After Transformation”.

·       Similarly section headers: Section 4.4 (Line 384) needs to be bolded.

·       Figure 3, 4, 5, 6, 7, 14, and 15 need better image quality. Please check and reupload.

·       Table 6. The footnote needs to be adjusted to “Transfer Learning (TL) based”.

6)     The conclusion section needs to be improved by including the limitations of the proposed approach and future research directions of this work.

Author Response

Reviewer #3

Notes on Revision

Re: Manuscript sensors-194-8894, entitled OHetTLAL: An Online Transfer Learning method for Fingerprint-based Indoor Positioning

We would like to express our gratitude to the editors and the anonymous reviewers for their constructive suggestions and criticism. The comments are well taken, and the manuscript has been revised accordingly. Below please find our responses to the reviewers’ comments. Also, for the reviewers’ convenience, major changes are written in YELLOW in the revised manuscript.

Responses to the Comments of Reviewer 3

We thank the Reviewer for the comments that helped us to improve the quality of the manuscript.

Comments to the Author

This paper proposes an online heterogeneous transfer learning  algorithm for IPS-based RSS fingerprinting to improve the positioning performance in the target domain. It is an important topic and the paper studies the concept clearly. However, there are several points need to be addressed to improve the quality of the manuscript.

Suggestions to improve the quality of the paper are provided below:

Comment #1: what number of APs and how are they deployed in the evaluation areas A and B?  Do you have 416 APs in approx 1400 square meter area?1460m2

Response #1: Thank you for your valuable suggestion.

Response #1: Thank you for your valuable comment. Two real-world experiments were carried out to evaluate our proposed algorithm. Experimental settings and datasets were first presented, and the overall performance of the classifiers are analyzed. Both feature spaces of scenarios were considered (before and after transformation). The general layout schema of the two experiments generating Dataset A and B are described below:

Dataset A: The experiment was conducted at UESTC (University of Electronic Science and Technology of China) as depicted in Fig. 2 and the 21st floor of the Innovation building was taken as the indoor environment site with 416 APs, consisting of 10 offices and one corridor covers an area of 1460 m2 and partitioned into 210 reference points (RP) which represent for the target’s location。 Each RP is numbered with a label .  and  be the number of offline and online instances collected during the offline training phases and the online testing phase respectively. The RSS of fingerprints at the lth RP can be represented as a matrix:  Smartphones were used for data collection at the deployment area. The 416 are number of the access points (APs) where initial position of the target or position of base stations (BS) are stored in fingerprint database. This are the feature spaces that can be used to map with the online testing RSS measurements.

Fig. 2 Experimental environments for dataset A showing the RP locations in blue circles.

Dataset B: Another real-office experiment at UESTC (University of Electronic Science and Technology of China) on the 21st floor of the Innovation Building was also conducted in different environmental settings to validate our results as shown in Fig. 3. It covers an area of 1460 m2 and partitioned into 175 reference points (RP). 9 Wi-Fi access points were sparsely deployed to guarantee that at least three access points are detectable at each RP.

Fig. 3 Experimental layout conducted in UESTC for generating Dataset B.

Moreover, Table 1 below provides the summary of the datasets.

Table 1. Descriptive of the two real-office experiment at UESTC

Dataset

Metrics

Area (m2)

UE

Base station (BS)

#BSs

#RPs

A

RSS

1460

Smartphone

APs

416

210

B

RSS

1460

Smartphone

APs

9

175

Comment #2: Title of the manuscript is a little misleading. It sounds like a review paper, whereas the authors propose an approach for indoor positioning. My suggestion is to update the title as “ An online transfer learning method for fingerprint-based indoor positioning” to make it more clear.

Response #2: Thank you for your valuable suggestion. Agreed.

Comment #3: Abstract needs to be more concise and extensive details should be removed. Currently, it contains a very detailed explanation of the feature space (line 26-31) which can be removed and elaborated more in the methodology if needed.

Response #4: Thank you for your valuable suggestion. In the revised version, the details are removed from the abstract and we have updated as:

In an indoor positioning system (IPS), transfer learning (TL) methods are commonly used to predict the location of mobile devices under the assumption that all training instances of the target domain are given in advance. However, this assumption has been criticized for its shortcomings in dealing with the problem of signal distribution variations, especially in a dynamic indoor environment. The reasons are: collecting a sufficient number of training instances is costly, the training instances may arrive online, the feature spaces of the target and source domains may be different, and negative knowledge may be transmitted in the case of a redundant source domain. In this work, we proposed an online heterogeneous transfer learning (OHetTLAL) algorithm for IPS-based RSS fingerprinting to improve the positioning performance in the target domain by fusing both source and target domain knowledge. The source domain was refined based on the target domain to avoid negative knowledge transfer. The co-occurrence measure of the feature spaces ( )  was used to derive the homogeneous new feature spaces, and the features with higher weight values were selected for training the classifier because they could positively affect the location prediction of the target. Thus, the objective function was minimized over the new feature spaces. Extensive experiments were conducted on two real-world scenarios of datasets, and the predictive power of the different modeling techniques were evaluated for predicting the location of a mobile device. Results have revealed that the proposed algorithm outperforms the state-of-the-art methods for fingerprint-based indoor positioning and is found robust to changing environments. Moreover, the proposed algorithm is more resilient to fluctuating environments and mitigates model’s overfitting problem.

Comment #4: Please mention several applications of indoor localization technologies in the building domain as it is highly valuable for building community (Line 57-58). Please refer to the highlighted building applications such as include emergency management, plug load controls, and HVAC controls.

Response #3: Thank you for your valuable suggestion. We found the indicated cites are so important reads and helped us to enrich the applications of indoor localization technologies in the building domain. In the revised version, we have modified our work by incorporating several a[plications in building community.

Indoor localisation for building emergency management Filippoupolitis, A., Oliff, W., & Loukas, G. (2016, December). Bluetooth low energy based occupancy detection for emergency management. In 2016 15th international conference on ubiquitous computing and communications and 2016 International Symposium on Cyberspace and Security (IUCC-CSS) (pp. 31-38). IEEE.

Indoor localisation for smart plug load control

Tekler, Zeynep Duygu, et al. "Plug-Mate: An IoT-based occupancy-driven plug load management system in smart buildings." Building and Environment 223 (2022): 109472. 

Indoor localisation for smart HVAC controls

Balaji, B., Xu, J., Nwokafor, A., Gupta, R. and Agarwal, Y., 2013, November. Sentinel: occupancy based HVAC actuation using existing WiFi infrastructure within commercial buildings. In Proceedings of the 11th ACM Conference on Embedded Networked Sensor Systems (pp. 1-14).

Comment #5: Under Section 2.1 Indoor Positioning Systems, the authors mentioning about the importance of Wi-Fi based fingerprinting as it is available in Wi-Fi-enabled devices, which is true and at the same time Bluetooth Low Energy (BLE)-based technologies are also available in mobile devices and became very popular over the recent years. I highly suggest authors to include reviewing several works using BLE-based localization technologies. Some of them are given below. Please review these works under this section to enrich the content.

 Response #4: Thank you for your valuable suggestion. In the revised version, we have updated Section 2.1 to enrich the applications of Wi-Fi based fingerprinting in relation to other popular network technologies applied for indoor positioning including BLE based technologies. We also pointed out the challenges associated with the indicated technologies and here is the updated one:

2.1 Indoor Positioning System

Recently, due to the increased demand for LBS and applications, several indoor localization methods have been proposed and implemented [1]-[3], [37], but they are not suitable for indoor localization due to some technical limitations and additional infrastructure investment costs [33]-[35], [50]. As in [20] indicated, building-dependent technologies include Wi-Fi, cellular, and Bluetooth, which use the building's infrastructure, and technologies such as RFID, UWB, infrared, ultrasound, Zigbee, VLC, and acoustic signals, which require their infrastructure. In contrast, image-based technologies and dead reckoning are classified as building-independent. Due to their low cost and ease of installation, indoor positioning systems based on wireless networks for building communities, particularly Wi-Fi and BLE as they may utilize the building's infrastructure, have attracted a lot of attention from researchers and industry [21-27]. In [22], have also described the feasibility of BLE deployment in addressing Plug load management systems to reduce the rising energy consumption of plug loads in commercial buildings through different load monitoring and control strategies. They have further discussed a novel IoT-based occupancy-driven plug load management system called Plug-Mate mainly relies on the users’: a) high-resolution occupancy information obtained through a nonintrusive indoor localization system, b) plug load type information inferred through an advanced plug load identification feature, and c) diverse control preferences through a personalized user interface. On the other side, the Authors described the fact that deployment locations of the BLE occupancy sensors are highly dependent on two main determinants: a) a constant power availability to operate the sensors and b) communication range of BLE signal causing sensors to be set 10-15 meters apart from each other and evenly distributed to ensure complete signal coverage [22].

Another study in [51], have proposed a scalable and less intrusive occupancy detection method-based on smartphones’ device using BLE network technology to perform zone-level occupancy localization, without the need for a mobile application. Results [51] revealed that despite the supervised ensemble model produced the best performance in terms of accuracy and macro average f1-score, the semi supervised clustering model demonstrated practical advantages as it was able to produce a reasonable performance, while using a fraction of the training data. This could justify the semi supervised clustering algorithms might be effective in occupant’s detection for limited labeled samples. Moreover, a non-intrusive occupancy monitoring approach which leverages on existing BLE technologies found in smartphone devices has been proposed to track the occupants’ movement patterns using BLE beacons [26]. The proposed approach does not require the installation of a mobile application but occupant’s MAC address of their Bluetooth-enabled smartphone devices [26]. Generally, BLE technology has significantly lower power consumption and much lower bit rates than Wi-Fi technology, and is ideal for regular short-distance data transmission [28]. Thus, BLE technology is being used or limited for short length of advertising packet. In addition to that, intrusive indoor positioning based BLE technology using user’s mobile device are becoming a more challenging for the scalability or feasibility of deployment due the concern for user’s privacy. Thus, privacy security protocol need be preserved to ensure that the BLE’s beacon packet broadcasting will not reveal one’s identity otherwise scalability of the technology will be negatively affected. This is consistent with the claim that have been discussed in [52], regardless indoor positioning based BLE network technology is gaining importance due to its ubiquitous nature, low cost and flexibility, but further scientific developments are limited for novel technologies such as BLE, due to the lack of open datasets and corresponding frameworks suitable to compare and evaluate specialized localization solutions.

However, Wi-Fi fingerprinting [33]-[37] is becoming one of the most commonly used methods for indoor localization due to the ubiquity of Wi-Fi infrastructure and the popularity of Wi-Fi-enabled mobile devices. An IPS for commercial buildings has been developed to provide fine-grained occupancy-based HVAC (Heating, Ventilation and Air Conditioning systems (HVAC)) actuation [23]. This system exploits the existing Wi-Fi network infrastructure and occupants' smartphones with Wi-Fi enabled. They have also discussed about how using the existing infrastructure for occupancy detection greatly lowers the cost and labor of implementation and maintenance. Furthermore, noisy Wi-Fi signals and metadata regarding the building's occupants, access points, and HVAC zone have also been employed to alleviate the difficulties in occupancy sensing, which could make it a cost-effective technique. Another method of Wi-Fi indoor localization based on SVM is proposed [53] to identify the floor using Wi-Fi signal of each floor and the method achieved an accuracy rate of 99.09% for 3D positioning [53]. An indoor localization system-based Wi-Fi fingerprint using spatial multi-points matching has also been proposed to estimate the user’s position [54]. An indoor location tracking technology-based Wi-Fi fingerprint technique [55] has been proposed to improve user’s location accuracy using mobile communication technology and location tracking technology. Additionally, for an indoor IoT application using a Bayesian network and a limited radio map, a reliable 3D indoor positioning system-based Wi-Fi RSS fingerprinting has been developed [56]. However, to counteract the impact of received signal variations caused by multipath and signal attenuations throughout various time periods, the radio map should be updated with in short time frame.

 This method does not use radio signal propagation geometry but requires data acquisition and a built-in radio map in the offline phase, although it has notable shortcomings in the dynamic indoor environment and requires enough labeled samples, making it labor-intensive and costly [57]. Moreover, relatively satisfactory accuracy can be achieved by this method, but [33]-[35] the database fingerprint must be up to date, otherwise, the location performance will be severely degraded. Some research works have been done to reduce the effort and time required to create the radio map. Most notably, these include crowd-sourcing [58] and simultaneous localization and mapping over Wi-Fi [59]. However, crowd-sourcing requires the active participation of users or achieves low accuracy, and the Wi-Fi simultaneous localization and mapping approach also suffers from the high computational cost. In [60], it was also described how to speed up computation and enhance system accuracy for indoor environments by using a clustering approach based on fingerprinting. Compared to the non-clustering approach, which had an average distance error of 3.4 m, the suggested technique obtained an average distance error of 2.4 m. Various methods have been proposed in the literature to address the dynamic indoor environment, which results in low localization accuracy due to variations in fingerprint patterns over time. These methods can be classified into four groups: probabilistic methods [38], [39], machine learning methods [40], [41], using the quality of fingerprints of various signal features (such as Channel State Information (CSI) [342], Phase of Arrival (PoA) [43], Time of Flight (ToF) [44], etc.), and derived signal features (Signal Strength Difference (SSD) [15] and fused group of fingerprints [45], [46]).

Although the proposed methods [15], [38]-[46] have improved the location accuracy, they also suffer from fluctuations in the signal distribution and are not robust to a dynamic indoor environment, mainly due to two limitations: i) the measurements depend on a single fingerprint and cannot represent the dynamic scenario of the indoor environment; ii) the need for a sufficient number of labeled samples is also very costly. Therefore, a hybrid positioning system (HPS) has been proposed for various seamless localization applications (Wi-Fi, Bluetooth, UWB, and ZigBee integrated into a hybrid base station (BS)) to solve the standalone positioning problem [8]. Other researchers in [47] have also proposed a hybrid indoor localization system based on the IMU sensor and smartphone camera, which has better positioning performance than the standalone system due to the possible combination of errors compensated by each technology. However, hybrid BS is not the subject of our work and is not economically feasible. On the other hand, computational complexity is a serious problem for hybrid indoor positioning systems. Research-based on machine learning algorithms (ML) have also been proposed to address the indoor location problem [40], [41], [61], [62]. A limited number of researchers have focused on reducing the calibration overhead for indoor WLAN locations [63], [64]. The impact of reducing the number of samples collected at each coverage point and reducing the number of coverages points on the accuracy of location determination has been studied [63], [64]. To compensate for the performance degradation with reduced calibration, both linear and kernel-based interpolation methods have been proposed [63], [64] to patch an incomplete radio map. The hidden Markov model [64] and hierarchical Bayesian model [65] have also been proposed to exploit the information contained in the unlabeled user traces and improve the accuracy of location estimation. A label propagation method [66], [67] has also been used to learn from labeled and unlabeled data under the assumption that similar data samples should have similar labels. However, this approach is not suitable for online prediction.

  1. C. Ng, P. Spachos and K. N. Plataniotis, "COVID-19 and Your Smartphone: BLE-BasedSmart Contact Tracing," in IEEE Systems Journal, vol. 15, no. 4, pp. 5367-5378, Dec. 2021, doi: 10.1109/JSYST.2021.3055675.

Tekler, Z.D., Low, R., Gunay, B., Andersen, R.K. and Blessing, L., 2020. A scalable Bluetooth Low Energy approach to identify occupancy patterns and profiles in office spaces. Building and Environment171, p.106681.

Baronti, Paolo, et al. "Indoor bluetooth low energy dataset for localization, tracking, occupancy, and social interaction." Sensors 18.12 (2018): 4462.

Filippoupolitis, A., Oliff, W. and Loukas, G., 2016, October. Occupancy detection for building emergency management using BLE beacons. In International Symposium on Computer and Information Sciences (pp. 233-240). Springer, Cham.

Tekler ZD, Low R, Blessing L. An alternative approach to monitor occupancy using bluetooth low energy technology in an office environment. InJournal of Physics: Conference Series 2019 Nov 1 (Vol. 1343, No. 1, p. 012116). IOP Publishing.

Huang, Ke, Ke He, and Xuecheng Du. "A hybrid method to improve the BLE-based indoor positioning in a dense bluetooth environment." Sensors 19.2 (2019): 424.

Comment #5: There are several formatting issues throughout the manuscript.

  • Section 4.1 (Line 320) and Section 4.1.1 Dataset A (Line 321) need to be bolded as they are section headers.
  • I would suggest merging Section 4.2 (Line 338), Section 4.3 (Line 357) and naming the section as “Distribution of Homogeneous Feature Spaces. Then, you can divide it into subsections such as “Before Transformation” and “After Transformation”.
  • Similarly section headers: Section 4.4 (Line 384) needs to be bolded.
  • Figure 3, 4, 5, 6, 7, 14, and 15 need better image quality. Please check and reupload.
  • Table 6. The footnote needs to be adjusted to “Transfer Learning (TL) based”.

 Response #5: Thank you for your valuable suggestion. We have updated it and replaced all the Figures with a better resolution.

Comment #6: The conclusion section needs to be improved by including the limitations of the proposed approach and future research directions of this work.

Response #6: Thank you for your valuable suggestion. In the revised version, we have updated it and pointed out the future research directions as:

In this paper, we propose OHetTLAL, a novel technique for indoor fingerprint-based positioning problems that can improve learning performance in the target domain through knowledge transfer from the source domain. To this end, we derived new feature spaces (on which the model is trained) based on the co-occurrence of RSS measurements from mobile devices in the two domains in order to capitalize on knowledge that could improve target location prediction. To derive the new feature spaces, the co-occurrence measure of the features ( ) was computed. The higher the value of co-occurrence between the two domains, the more detected Wi-Fi APs are shared by these two domains and the more likely these source domains are related to the target domain, which positively affects the prediction of the target. The proposed objective function was minimized across the new feature spaces so that the positive knowledge of the related source domain received a higher weight and is transferred to the target domain for predicting a new mobile device. Experimental results show that the proposed algorithm was the best algorithm for fingerprint-based indoor positioning with the minimum root mean square error in all cases of the two real office scenarios. On the other hand, the Gaussian mixture model (GMM), had the highest mis-localization error rate. Our research shows that in OHetTL, the construction of new feature spaces is critical for extracting relevant information for effective target location prediction. It should be noted that these experimental results may differ in other test phase environments.

The effects of different dimensions of feature spaces used for training indoor positioning classifiers have also been evaluated using two real life datasets. The proposed algorithm has succeeded in improving the location accuracy by transferring knowledge from the source domain to the target domain for both datasets after new feature spaces are derived based on the weight co-occurrence of both instances. This difference leads us to an important finding, namely that all classifiers achieved lower RMSE both before and after transfer learning over the new feature spaces of dimension 9. However, there is no strong evidence that negative knowledge was not transferred from the source to the target domain. In this case, especially for the scenario of dataset A with 9 feature spaces, the measurements coming from all Wi-Fi access points were not refined or reviewed to determine whether or not they should be used for location analysis because they could negatively affect the positioning process. However, the contributions of these features to target detection were not as significant (since their co-occurrence weight scores of those three features were almost negligible) as for the feature spaces with a dimension of 6. This could justify that a model with a lower RMSE does not mean that it is always the most consistent model, but that we need to examine the parameters for their negative effects on target prediction. Our results show, however, that the proposed algorithm is more resilient to fluctuating environments and mitigates the problem of overfitting the model. Furthermore, the OHetTLAL-based fusion of both knowledge domains could significantly improve positioning performance without any extra cost associated to offline calibrations effort.

Round 2

Reviewer 1 Report

Thanks for addressing my comments, but there are still a few minor corrections as follows:

- Is there any justification why when increasing the number APs in Figures 14,15 and 23, some algorithms achieved higher RMSE, i.e MLP? Please write a valid justification. 

- The authors should mention in the conclusion the localization error achieved by their proposed algorithm for the two datasets. It is very important to show the readers the achieved system accuracy. 

Author Response

Reviewer #1:

Notes on Revision

Re: Manuscript sensors-194-8894, entitled OHetTLAL: An Online Transfer Learning method for Fingerprint-based Indoor Positioning

We would like to express our gratitude to the editors and the anonymous reviewers for their constructive suggestions and criticism. The comments are well taken, and the manuscript has been revised accordingly. Below please find our responses to the reviewers’ comments. Also, for the reviewers’ convenience, major changes are written in YELLOW in the revised manuscript.

Responses to the Comments of Reviewer 1

We thank the Reviewer for the comments, suggestions and questions that helped us to improve the quality of the manuscript.

#Round 2

Comments and Suggestions for Authors

Thanks for addressing my comments, but there are still a few minor corrections as follows:

Comment #1: Is there any justification why when increasing the number APs in Figures 14,15 and 23, some algorithms achieved higher RMSE, i.e MLP? Please write a valid justification. 

Response #1: Thank you for your suggestion.

The original dataset A was with dimensions of 416. However, when we look the measurements weights on modeling the variations of the positioning performance using data processing technique and scrutinized critically; we found that some of the features are not significant for the predictive purpose. Besides, some of the features are nor relevant for the modeling purpose. Therefore, we also proposed a new metrics to derive a new feature spaces based on the co-occurrence of both instances of training and testing datasets. Moreover, based on the exclusion criteria’s we set, we derived a new refined feature spaces with dimensions of 41 and 15 as two possible scenarios for comparison purpose. Accordingly, Fig. 14 confirms that the proposed algorithm is the best algorithm with the lowest root mean squared error for dataset A with 41 feature spaces. In this case, the measurements collected from the remaining Wi-Fi access points were not included in the analysis because they could negatively affect the prediction task as the coincidence weights of these Wi-Fi access points from both domains of instances were found to be insignificant, so including them would spuriously increase the positioning accuracy. On the other hand, Fig. 15 illustrates that the proposed algorithm was also found to be as the best algorithm with the lowest root mean squared error for dataset A with 15 feature spaces. In this case, the measurements that came from the 26 Wi-Fi access points were also excluded from the analysis because the co-occurrence weight scores of these access points from both domains of the instances were insignificant, so including them would falsely inflate our positioning accuracy. Similarly, all classifiers achieved a minimal RMSE after transfer learning was applied, although slight differences were observed as some instances were able to falsely improve the RMSE. This indicates that our proposed algorithm succeeded in improving the positioning accuracy by transferring knowledge from related source domains to the target domain. It can be also observed that positive knowledge from the source domain was transferred to the prediction task.

Moreover, dataset B also coincides with our hypothesis that the number of original feature spaces considered is 9 later reduced to 6 feature spaces based on the weight co-occurrence measures. The positioning accuracy of the proposed model is found to be the best with minimum root mean square error. It is not necessarily the positioning performance to improve as number of feature spaces increases instead how the features are important in capturing the maximum variation of the model is the main determinant.

In summary, the positioning performance as per our dataset revealed that it is highly important to extract the most significant predictors that could capture the maximum variance of the model using some metrics in order to evaluate the significance of each Wi-Fi AP deployment. Thus, we discovered that not all available feature spaces need be considered for positioning estimation for at least four reasons: (a) some features may be irrelevant and result in fingerprint duplication or pattern fingerprint mismatch; (b) large feature spaces necessitate a massive deployment of Wi-Fi access points, which is prohibitively expensive from a cost standpoint; (c) from a technical standpoint, computational cost and memory usage are quite expensive when dealing with higher dimensions of features, and (d) model over-fitting is also a serious issue for higher dimensions of feature spaces.

Comment #2: The authors should mention in the conclusion the localization error achieved by their proposed algorithm for the two datasets. It is very important to show the readers the achieved system accuracy. 

Response #2: Thank you for your suggestions. In the revised version, we have added the following ideas as below:

Results also analyzed the impact of different feature spaces used to train indoor positioning classifiers using dataset A with dimensions of 41 and 15, respectively. It was found that the proposed algorithm has the lowest root mean square error, i.e., 32% and 25% less error in positioning the mobile device for both scenarios. Similarly, the effects of different dimensions of feature spaces used for training indoor positioning classifiers, using dataset B with a dimension of 9 and 6, respectively. It was found that the proposed algorithm has the lowest root mean square error, i.e., 33% and 22% less mis localizations of mobile devices for both scenarios. This indicates that the proposed algorithm has succeeded in improving the positioning accuracy by transferring knowledge from the source domain to the target domain. It can be observed that not only positive knowledge from the source domain was transferred to the prediction task, but also the computational cost and memory consumption were effectively improved by excluding irrelevant features from the analysis. The difference in positioning accuracy obtained due to the feature spaces between the two scenarios is about 7% and 11% for both datasets of A and B respectively.

Reviewer 3 Report

Thank you for addressing my comments and concerns carefully. The current version of the manuscript is ready for publication. Great work!

Author Response

Reviewer #3

Notes on Revision

Re: Manuscript sensors-194-8894, entitled OHetTLAL: An Online Transfer Learning method for Fingerprint-based Indoor Positioning

We would like to express our gratitude to the editors and the anonymous reviewers for their constructive suggestions and criticism. The comments are well taken, and the manuscript has been revised accordingly. Below please find our responses to the reviewers’ comments. Also, for the reviewers’ convenience, major changes are written in YELLOW in the revised manuscript.

Responses to the Comments of Reviewer 3

We thank the Reviewer for the comments that helped us to improve the quality of the manuscript.

Comments to the Author

Comments and Suggestions for Authors

Thank you for addressing my comments and concerns carefully. The current version of the manuscript is ready for publication. Great work!

Thank you so much for your comments and suggestions; it has greatly enhanced our work.